# Phosphoproteomic identification of ULK substrates reveals VPS15-dependent ULK/VPS34 interplay in the regulation of autophagy

Thomas John Mercer[1], Yohei Ohashi[2], Stefan Boeing[3], Harold B J Jefferies[1], Stefano De Tito[1,4], Helen Flynn[4] (iD), Shirley Tremel[2], Wenxin Zhang[1], Martina Wirth[1], David Frith[5], Ambrosius P Snijders[5], Roger Lee Williams[2] & Sharon A Tooze[1,*] (iD)

## Abstract

Autophagy is a process through which intracellular cargoes are catabolised inside lysosomes. It involves the formation of autophagosomes initiated by the serine/threonine kinase ULK and class III PI3 kinase VPS34 complexes. Here, unbiased phosphoproteomics screens in mouse embryonic fibroblasts deleted for Ulk1/2 reveal that ULK loss significantly alters the phosphoproteome, with novel high confidence substrates identified including VPS34 complex member VPS15 and AMPK complex subunit PRKAG2. We identify six ULK-dependent phosphorylation sites on VPS15, mutation of which reduces autophagosome formation in cells and VPS34 activity *in vitro*. Mutation of serine 861, the major VPS15 phosphosite, decreases both autophagy initiation and autophagic flux. Analysis of VPS15 knockout cells reveals two novel ULK-dependent phenotypes downstream of VPS15 removal that can be partially recapitulated by chronic VPS34 inhibition, starvation-independent accumulation of ULK substrates and kinase activity-regulated recruitment of autophagy proteins to ubiquitin-positive structures.

**Keywords** p62; PIK3R4; PRKAG2; ULK1; VPS15

**Subject Categories** Autophagy & Cell Death; Post-translational Modifications & Proteolysis

The EMBO Journal (2021) 40: e105985

## Introduction

Autophagy maintains cellular homeostasis by allowing removal of cytotoxic elements such as damaged organelles, protein aggregates and intracellular pathogens, and its implication in both physiology and pathophysiology has rapidly broadened in recent years.

Multiple subtypes of autophagy exist; however, the best characterised is macroautophagy (herein autophagy), which involves the sequestration of cytosolic components into double-membraned vesicles, termed autophagosomes, which fuse with lysosomes leading to the degradation of their components.

Controlled by upstream nutrient and energy-sensing kinases (mTORC1 and AMPK), the autophagy initiating ULK kinase complex becomes active and translocates to the ER (Ganley *et al*, 2009; Hosokawa *et al*, 2009; Kim *et al*, 2011; Karanasios *et al*, 2013). Here, association of ATG9A vesicles leads to production of phosphatidylinositol-4-phosphate (PI4P) and recruitment of lipid kinase VPS34 complex I leading to the formation of a phosphatidylinositol-3-phosphate (PI3P)-enriched domain termed the omegasome (Karanasios *et al*, 2013; Karanasios *et al*, 2016; Judith *et al*, 2019). VPS34 activity stabilises ULK1 at omegasomes (Karanasios *et al*, 2013) and via the recruitment of PI3P-binding proteins such as WIPI2b drives formation of double-membraned phagophores. WIPI2b-localised lipidation of Atg8p homologs (LC3/GABARAP-family proteins) (Dooley *et al*, 2014) and the active transport of lipids via ATG2A/B (Chowdhury *et al*, 2018; Osawa *et al*, 2019; Valverde *et al*, 2019) allow phagophore expansion and engulfment of bulk and selective cargoes, likely facilitated by the lipid scramblase activity of ATG9A (Maeda *et al*, 2020). Expansion is further promoted by re-recruitment of ULK complex to phagophores by Atg8p homologs (Kraft *et al*, 2012; Joachim *et al*, 2015).

The pentameric ULK complex contains the serine/threonine kinase ULK1 or close homolog ULK2 (collectively ULK), ATG13, ATG101 and dimeric FIP200 (Shi *et al*, 2020), and is the sole protein kinase in the autophagic signalling cascade. It integrates multiple stimuli including nutrient deprivation, genotoxic stress (Torii *et al*, 2016; Torii *et al*, 2020), mitochondrial dysfunction (Egan *et al*, 2011; Tian *et al*, 2015; Vargas *et al*, 2019) and intracellular pathogen presence (Ravenhill *et al*, 2019) to induce autophagy via direct substrate phosphorylation as well as via non-catalytic scaffolding

1  Molecular Cell Biology of Autophagy, The Francis Crick Institute, London, UK
2  MRC Laboratory of Molecular Biology, Cambridge, UK
3  Bioinformatics and Biostatistics, The Francis Crick Institute, London, UK
4  Institute of Experimental Endocrinology and Oncology (IEOS), National Research Council, Naples, Italy
5  Proteomics, The Francis Crick Institute, London, UK
   *Corresponding author (lead contact). Tel: +44 20 37961340; E-mail: sharon.tooze@crick.ac.uk

functions (Mercer *et al*, 2018). Whilst ULK signalling is primarily understood as crucial for autophagy initiation, recent insights have established it as a multistage regulator of autophagy (Chan *et al*, 2009; Itakura & Mizushima, 2010; Egan *et al*, 2015; Petherick *et al*, 2015; Wang *et al*, 2018b). Furthermore, whilst ULK double knock-out (DKO) mice display perinatal lethality observed in ATG knock-out animals (McAlpine *et al*, 2013; Cheong *et al*, 2014; Lechauve *et al*, 2019), they also have distinct phenotypes such as increased embryonic mortality, structural abnormalities in the lung (Cheong *et al*, 2014) and defects in both axon guidance (Wang *et al*, 2018a) and erythrocyte ROS neutralisation (Li *et al*, 2016). Mutation of *ULK1* has also been implicated in various cancers (Kumar & Papaleo, 2020), and in cell-based models, ULK signalling has been implicated in necroptosis (Wu *et al*, 2020), ER-Golgi traffic (Joo *et al*, 2016) and stress granule clearance (Wang *et al*, 2019).

Despite the pivotal roles of ULK in autophagy as well as diverse physiological pathways, the mechanisms by which it functions is unclear. It is known however that ULK kinase activity is crucial for autophagy (Hara *et al*, 2008; Chan *et al*, 2009; Egan *et al*, 2015; Petherick *et al*, 2015) and the autophagic machinery contains several ULK substrates (Mercer *et al*, 2018), including multiple components of the VPS34 complex I.

The class III lipid kinase VPS34 constitutes the catalytic core of complex I and II (Itakura *et al*, 2008; Kim *et al*, 2013). VPS34 binds the pseudokinase VPS15 (also known as p150/PIK3R4) greatly increasing VPS34 activity *in vitro* (Yan *et al*, 2009). The activity and stability of the VPS34-VPS15 subcomplex is further augmented by association with Beclin1 and either ATG14 or UVRAG, forming VPS34 complex I (CI) and II (CII), respectively, (Yan *et al*, 2009; Kim *et al*, 2013; Rostislavleva *et al*, 2015). VPS34 CI contains a 5[th] component, NRBF2, which facilitates dimerisation of the pentameric complex, and promotes its stability and kinase activity (Ohashi *et al*, 2016; Ma *et al*, 2017; Young *et al*, 2019). Amongst other functions, VPS34 CII regulates endolysosomal sorting (Backer, 2016; Jaber *et al*, 2016), autophagosome-lysosome fusion (Sun *et al*, 2011; Kim *et al*, 2015) and autophagosome-lysosome reformation (Munson *et al*, 2015), whilst VPS34 CI is crucial for autophagosome biogenesis (Brier *et al*, 2019). Furthermore, VPS34 CI-dependent autophagosome-lysosome fusion has been reported (Hegedűs *et al*, 2016; Takáts *et al*, 2020).

Protein phosphorylation is implicated in the regulation of VPS34 activity by nutrient status. Of the 6 VPS34 complex components (VPS34, VPS15, Beclin1, UVRAG, ATG14 and NRBF2), 3 are substrates of mTOR (UVRAG, ATG14 and NRBF2) (Yuan *et al*, 2013; Kim *et al*, 2015; Munson *et al*, 2015; Ma *et al*, 2017) and 4 are targeted by ULK1 (VPS34, Beclin1, ATG14 and NRBF2) (Yuan *et al*, 2013; Egan *et al*, 2015; Park *et al*, 2016; Ma *et al*, 2017; Mercer *et al*, 2018; Park *et al*, 2018; Birgisdottir *et al*, 2019), with a variety of mechanistic consequences described. No functional phosphorylation sites on VPS15 have been annotated to date.

Being one of the few autophagy proteins that can be targeted pharmacologically (Egan *et al*, 2015; Petherick *et al*, 2015), understanding ULK's substrate repertoire is essential for the development of therapies to manipulate autophagy. In 2015, Egan *et al* (2015) published a consensus motif for ULK1 which was used to screen for novel substrates. However, many substrates published independently match the motif poorly (Mercer *et al*, 2018), and the use of the motif in the screen along with both a focus on components of

the early autophagic machinery and the reliance on kinase/substrate overexpression limited the identification of physiologically relevant substrates. We analysed phosphoproteomes of wild-type and ULK DKO cells cultured under autophagy-inducing conditions to obtain a high confidence shortlist of substrates. Assessing direct phosphorylation by ULK *in vitro* validated several new substrates, including components of both the VPS34 (VPS15 and UVRAG) and AMPK (PRKAG2 and PRKAB2) complexes. We reveal the complexity of ULK signalling by describing the upstream stimuli for ULK-dependent phosphorylation of PRKAG2 and VPS34, including nutrient, energy and iron homeostasis.

VPS15 is phosphorylated at six sites by ULK and mutation of these residues inhibits autophagy in cells and VPS34 lipid kinase activity *in vitro*. Mutation of the endogenous VPS15 gene supported the importance of the pseudokinase domain for VPS15's biological activity. VPS15 KO clones accumulated ULK phospho-substrates and aberrant structures containing autophagy proteins. These structures could be partially recapitulated in unmodified cells by treatment with VPS34 inhibitors. Intriguingly, the recruitment and distribution of a subset of autophagy proteins at these structures was controlled by ULK activity status. Together, our identification of new ULK substrates and the characterisation of the ULK-VPS15 signalling axis illuminate novel functions of this deeply conserved kinase.

## Results

### The phosphoproteome is significantly altered in the absence of ULK

To identify novel ULK substrates, wild-type (WT) and Ulk1$^{-/-}$ Ulk2$^{-/-}$ (double knockout—DKO) mouse embryonic fibroblasts (MEFs) were analysed by SILAC-based quantitative phosphoproteomics (Fig 1A).

Three independent experiments were performed utilising 2 WT and DKO MEF types and 2 autophagy-inducing conditions: (i) (SV40/Torin 1); (ii) (SV40/starvation) and (iii) (spontaneously immortalised/starvation). This experiment revealed that the phosphoproteome is significantly altered in the absence of the ULK kinases. The combined data set contained 17,940 unique phosphorylation events detected in both forward and reverse data sets, amongst which 512 phosphorylation sites were highly depleted in DKO cells (Table EV1). Reproducibility was higher within experiments than between experiments (Fig EV1A).

Similar numbers of phosphopeptides appeared to be enriched as well as depleted in the DKOs, and when the distribution of phosphopeptides matching the reported ULK1 consensus motif was examined, no significant enrichment was observed in either MEF cell line (Fig 1B). Notably, 4 of the 9 phosphorylated residues in Fip200 (gene name *Rb1cc1*), a known ULK substrate, were depleted in DKOs, 1 of which matched the consensus motif (Fig 1B). Consensus sequence enrichment was used to identify the kinase(s) responsible for the phosphopeptides depleted in DKOs; however, the resulting motif was strikingly dissimilar to the published ULK1 motif (Egan *et al*, 2015) indicating that the majority of phosphosites depleted were not direct ULK substrates (Fig 1C).

We then interrogated the data using additional information. As ULK preferentially phosphorylates proteins that it stably interacts

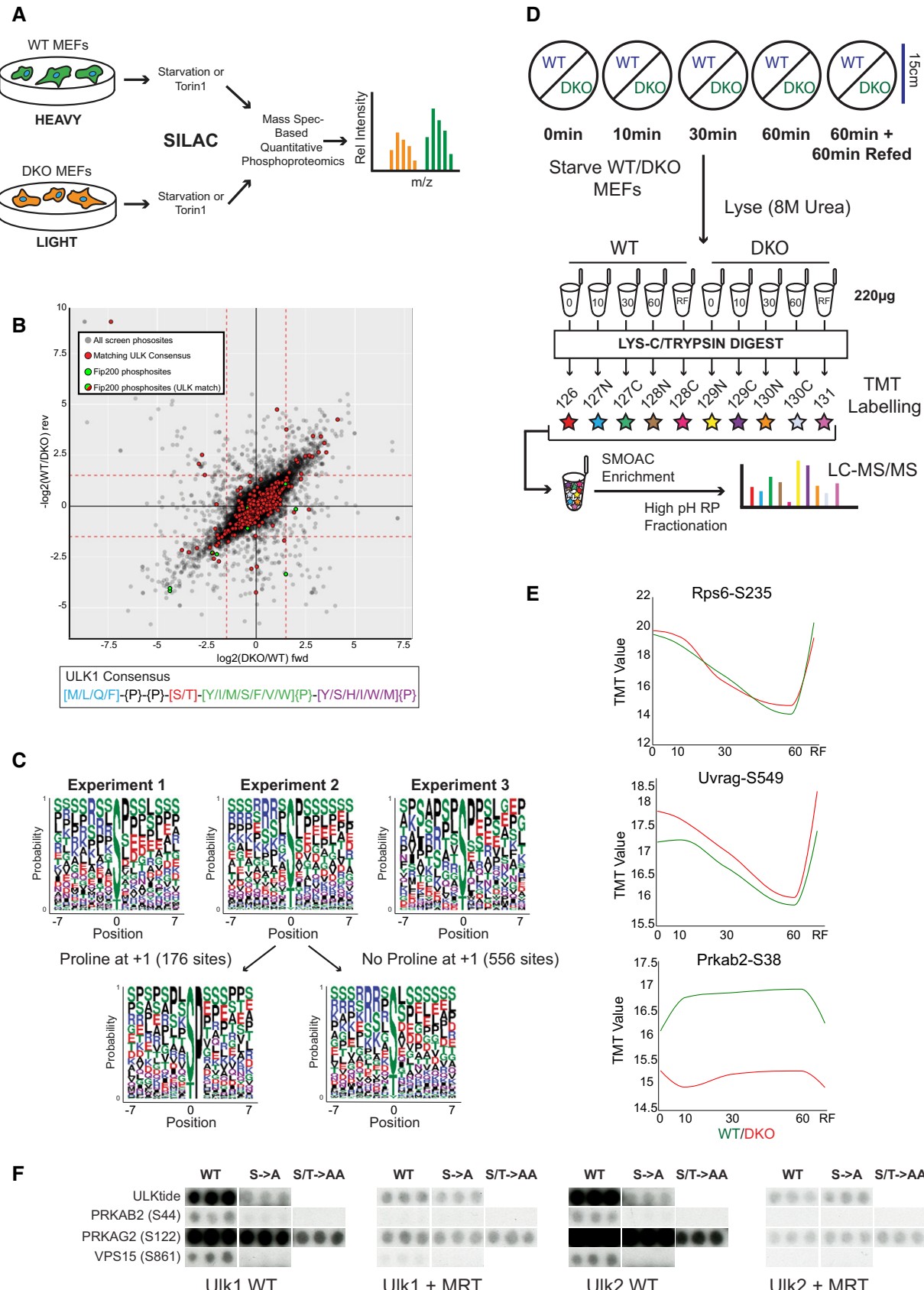

**Figure 1.**

**Figure 1.  SILAC- and TMT-based phosphoproteomics coupled with peptide array-based in vitro kinase assays to identify ULK substrates.**

A   SILAC screen design.

B   Scatterplot with relative phosphopeptide enrichment in Ulk1/2 double knockout (DKO) vs wild-type (WT) MEFs in experiment 1. Log$_2$ SILAC ratios from forward (fwd) and reverse (rev) conditions are plotted, with reciprocal ratios shown for the reverse condition. 16,308 phosphosites were detected in the forward and reverse experiments, and those with a log$_2$ SILAC ratio of $< -1.5$ in both (lower left quadrant) were considered to be depleted in DKOs. Nine phosphopeptides were assigned to the Fip200 (gene name *Rb1cc1*), shown in green, and 560 matched the ULK1 motif sequence shown in red. Two Fip200 phosphopeptides matched the ULK1 motif, indicated in red and green, 1 of which was significantly depleted in DKOs. See Materials and Methods for use of ULK1 motif.

C   Relative enrichment of amino acids centred on phosphosites depleted in DKOs depicted as consensus logos from experiments 1, 2 and 3. For experiment 2, average motifs for peptide subsets with and without proline at position +1 are shown in separate logos to reveal the contribution of proline-directed kinases.

D   TMT starvation time course protocol.

E   Profile plots for the 3 control phosphopeptides. TMT enrichment (Y axis, log$_{10}$ intensity) from the starvation time course (X axis, mins; RF = Refed) with WT enrichment values in green and DKO values in red. In the refed condition full media was added after 60 min of starvation and cells harvested 60 min later.

F   Control and experimental peptides selected from peptide array-based *in vitro* kinase assay (see Fig EV2 for full details).

Source data are available online for this figure.

with (Mercer *et al*, 2018), we generated a protein-protein interaction network (PPI) predominantly based on experiment-derived interaction databases (Türei *et al*, 2015; Huttlin *et al*, 2017; Giurgiu *et al*, 2019). A number of phosphopeptides reproducibly depleted in the DKOs mapped to proteins within 2 nodes of ULK1 and ULK2 (Fig EV1B). These include the AMPK complex subunit Prkab2, for which significant dephosphorylation was detected at S38, a previously identified *in vitro* substrate of Ulk1 (Löffler *et al*, 2011).

The SILAC data set is a rich resource for phosphopeptides depleted in ULK DKOs, however, due to the inability to measure phosphorylation in basal conditions, the false positive identification rate could not be ascertained. Exemplifying this, for some of the selected substrates, most if not all of the unique phosphopeptides identified in the data set were depleted in DKOs. In these cases, the peptide sequences surrounding the phosphoacceptor were often highly divergent and therefore unlikely to be targeted by ULK (e.g. see Sorbs2 in Table EV1); thus, their decrease was potentially indicative of protein level variation rather than loss of ULK-dependent phosphorylation. To reduce the impact of false positives, a second ULK substrate screen was employed in which phosphorylation was measured over a starvation time course.

## Time-resolved quantitative phosphoproteomics identifies ULK substrates

Populations of WT and DKO MEFs were starved of amino acids and serum for 0, 10, 30 or 60 min, or starved for 60 min and then refed for 60 min (RF) before lysis and mass spectrometry analysis using tandem mass tag (TMT) labelling (Fig 1D). After removing

sites not detected in all 10 conditions, around 15,000 unique phosphopeptides remained in the TMT data set. Unbiased correlation cluster analyses revealed that both starvation- and ULK-dependent changes were readily detectable in the data set (Fig EV1C). Positive controls for nutrient-regulated phosphorylation include multiple substrates downstream of ULK's primary regulator mTORC1 (Mercer *et al*, 2018), such as the indirect substrate Rps6 S235 (Rosner *et al*, 2011) and direct substrate Uvrag S549, implicated in autophagic lysosome reformation (Munson *et al*, 2015) (Fig 1E). Furthermore, the ULK substrate Prkab2 S38 (Löffler *et al*, 2011) was phosphorylated upon starvation in WT MEFs whilst phosphopeptide levels plateaued at a lower baseline level in DKOs (Fig 1E). Around 1/5$^{th}$ of the sites reproducibly depleted in the SILAC analysis (Table EV1) were detected in the TMT screen, with the averaged profile indicative of basal depletion in DKOs (Fig EV1D). Some of these 100 sites displayed very little variation in all conditions tested indicating that the correlation between SILAC and TMT screens was not perfect (Fig EV1E). These insights supported that many of the DKO MEF phosphopeptides depleted in the SILAC screen were not direct ULK substrates, validating our use of confirmatory screening in substrate identification.

To identify novel ULK substrates, three comparisons were chosen to describe both the starvation and the ULK dependence of the phosphorylation events in the TMT data set. In each case, the difference in phosphopeptide enrichment between 2 specified conditions was considered (Fig EV1F), with statistical similarity to Prkab2 S38 used as a fourth variable (see Materials and Methods for notes on TMT analysis). Phosphopeptides satisfying all variables were shortlisted as putative ULK substrates. The SILAC and TMT data sets were then

**Figure 2.  PRKAG2 S124 is directly phosphorylated by ULK and AMPK and regulated by serum status.**

A   Immunopurified C-terminally GFP-tagged PRKAG2 N-terminal tail fragments (amino acids 1–136, 1–242) with deletion of Δ122–124 or without (WT) were incubated with full-length Ulk1 wild type (WT) or kinase inactive (KI) for *in vitro* kinase assays. Autoradiograms and Coomassie gels reveal $^{32}$P incorporation and protein level, respectively.

B   GFP-tagged PRKAG2 N-terminal tail fragments (amino acids 1–242) were immunopurified and phosphorylated *in vitro* by Ulk1 WT or KI. Immunoblot analysis revealed that S124 is phosphorylated by Ulk1 *in vitro*.

C   Full-length FLAG-PRKAG2 WT and Δ124 were co-expressed with full-length Ulk1 WT and incubated in the presence of 1 μM MRT68921. Western blot analysis revealed that Ulk1 overexpression strongly promotes S124 phosphorylation in cells.

D   Cells expressing empty vector (EV), or PRKAA1-myc, PRKAB2 and PRKAG2-FLAG (AMPK) alone or with myc-Ulk1 WT (ULK) were cultured in full medium (F), EBSS (St) or serum-free medium (SFM) alone or in the presence of 991 (1 μM) or MRT68921 (1 μM) for 90 min as indicated. 2% Loading control and pulldown samples were analysed by Western blot.

Data information: (B, C) Identical samples were analysed on separate blots, separated by dashed lines.

Source data are available online for this figure.

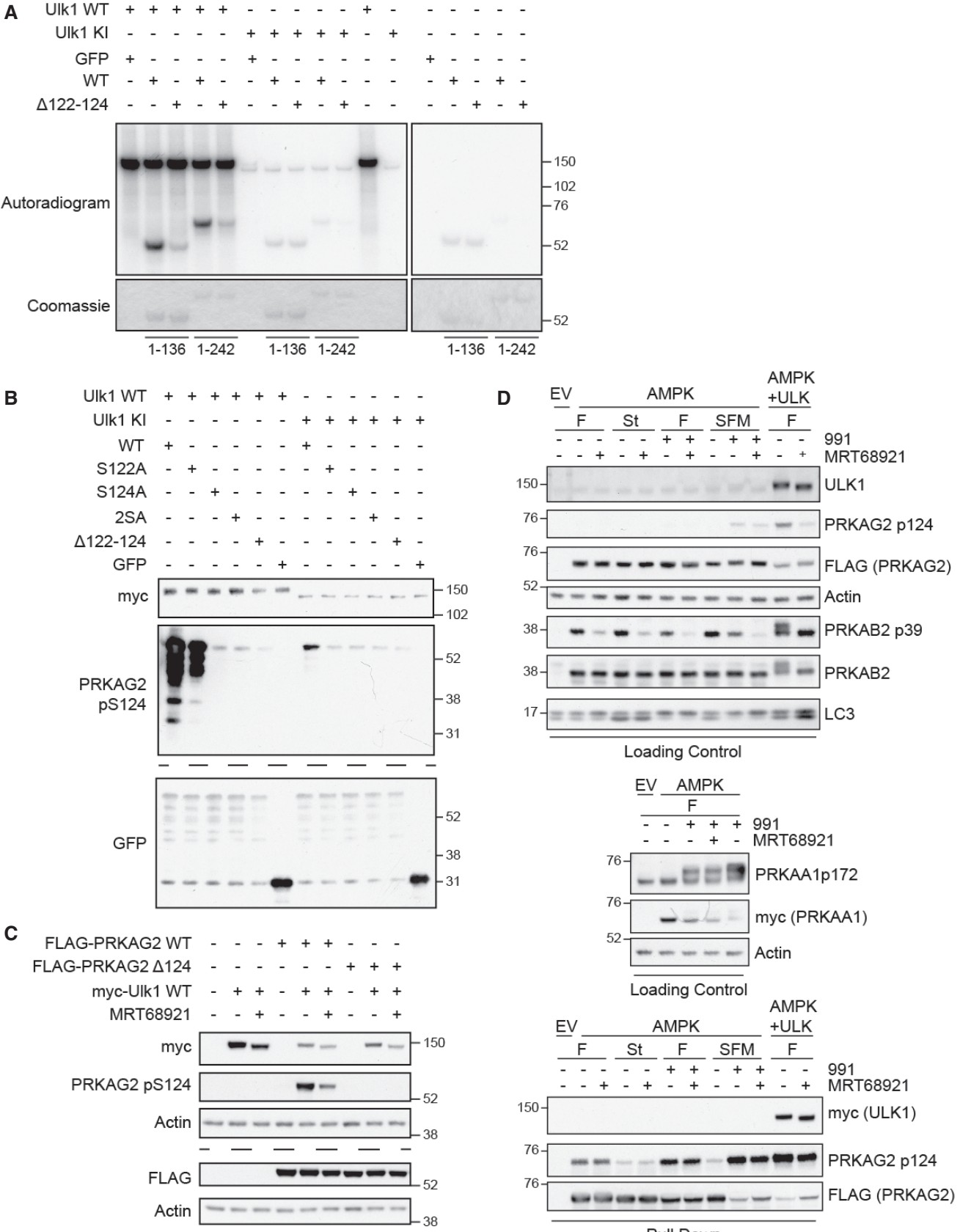

**Figure 2.**

cross-referenced to generate a high confidence list of putative substrates, to which three phosphopeptides detected in the SILAC screen only were included based on biological interest. Finally, phosphopeptides with particularly low conservation in *H. sapiens* in the 15mer peptide spanning the phosphoacceptor were removed to focus on substrates with relevance in humans, resulting in the high confidence shortlist (Appendix Table S1).

### Triaging shortlisted substrates

ULK remains active after purification from cell lysates, and *in vitro* phosphorylation has been used to identify substrates such as Prkab2 S38 (Löffler *et al*, 2011). Furthermore, the ULK1 motif itself was calculated by measuring the efficiency with which ULK1 phosphorylated a positional scanning peptide library (Egan *et al*, 2015). Therefore, phosphorylation by Ulk1 *in vitro* was used to triage the high confidence shortlist.

For each shortlisted site, 15mer peptides were arrayed in triplicate. Up to 4 peptide variants were tested, with WT peptides included alongside single phosphomutants of the identified phosphoacceptor site (S->A). If present, additional serines or threonines were changed to alanine in total phosphomutant (S/T->AA) peptides. If the 15mer was divergent between human and mouse, the murine peptide sequence was included. Finally, a number of validated substrates were also included as positive controls (Mercer *et al*, 2018): ATG13 S355 (S318 in isoforms 2 and 3), ATG14 S29, Beclin1 S15, NR3C2 S843, PRKAB2 S39, ULK1 S1042/T1046 and ULKtide (closely matching the ULK consensus signature; Egan *et al*, 2015). See Appendix Table S2 for peptide identities.

Ulk1 or Ulk2 complexes were prepared in HEK293A and used to phosphorylate arrays *in vitro* (Fig EV2A and B). Incubation with a small molecule ULK inhibitor (MRT68921; Petherick *et al*, 2015) prevented array phosphorylation, validating direct targeting by ULK (Fig EV2B).

Annotation of the WT-phosphorylated peptide arrays (Fig EV2B) revealed the potential ULK substrates (Fig EV2C and Appendix Table S2). These included components of the VPS34 and AMPK complexes (Fig 1F).

### The AMPK component PRKAG2 is a novel ULK substrate

SILAC data suggested that the AMPK complex subunit PRKAG2 is phosphorylated by ULK at S122 (Table EV1); however, co-mutation of S117 and S124 was required to prevent Ulk1 phosphorylation *in vitro* (Figs 1F and EV2C). PRKAG2 consists of 4 C-terminal nucleotide-binding CBS (cystathionine β-synthase) domains found in all human PRKAG homologs (PRKAG1/2/3) and an unstructured N-terminal tail (amino acids 1–242) that regulates subcellular localisation (Pinter *et al*, 2013; Cao *et al*, 2017). Mutations in *PRKAG2* are implicated in hypertrophic cardiomyopathy (PRKAG2 syndrome (Pöyhönen *et al*, 2015)) with a disease-causing mutation (G100S) close to the predicted ULK target sites (Zhang *et al*, 2013). We therefore asked if this disease-relevant protein is phosphorylated by ULK.

As ULK phosphoacceptors have been identified in PRKAG1, which is mostly composed of the CBS domains and is highly similar to PRKAG2's C-terminus (Löffler *et al*, 2011), we used N-terminal tail fragments to focus on the potential substrates. When WT and Δ122–124 deletion mutants were used as substrates, only the WT

protein was phosphorylated by Ulk1 *in vitro* (Fig 2A). As mutation of S122 alone was ineffective in preventing phosphorylation by ULK *in vitro* (Fig 1F), and as S124 matched the ULK1 consensus motif, we speculated that S124 might be the major ULK phosphoacceptor in PRKAG2. Phosphospecific antibodies to PRKAG2 pS124 confirmed phosphorylation by Ulk1 *in vitro* (Fig 2B) and in cells (Fig 2C). Intriguingly, we discovered that S124 is likely also subject to AMPK-dependent autophosphorylation and that phosphorylation is highly sensitive to serum status, as serum removal decreased basal phosphorylation but increased autophosphorylation (Fig 2D). To our knowledge, PRKAG2 S124 is the first reported dual substrate of both AMPK and ULK.

These data support our approach in using unbiased phosphoproteomics followed by a confirmatory peptide array-based *in vitro* kinase assays to identify novel ULK substrates. We then focussed on a second candidate, VPS15.

### ULK phosphorylates VPS34 complex subunit VPS15

The pseudokinase VPS15 is best understood as a scaffolding subunit for VPS34 CI and CII. Whilst its importance has been revealed via knockout studies in multiple model organisms (Herman *et al*, 1991; Lindmo *et al*, 2008; Xu *et al*, 2011; Nemazanyy *et al*, 2013; Voigt *et al*, 2014; Anding & Baehrecke, 2015; Gstrein *et al*, 2018), it is the least studied VPS34 complex component.

Screen data indicated that ULK phosphorylates VPS15 at S861 (Figs 1F and EV2C) and S865 (Table EV1). We sought to corroborate that ULK kinases could phosphorylate VPS15 in cells. Co-expression of WT Ulk1 or Ulk2 with VPS15 led to a band shift in VPS15 (Fig 3A), indicative of multisite phosphorylation. To discover further phosphoacceptors, VPS15 was immunopurified from cells co-expressing either Ulk1 WT, Ulk1 KI (Kinase Inactive) or empty vector before mass spectrometry analysis, identifying 4 additional sites: S813, S879, S1039 and S1289 (Fig 3B). In total, 4 of the 6 putative ULK target sites were identified previously in phosphoproteomic screens (Hornbeck *et al*, 2015) and most matched the consensus motif to some degree (Fig 3C).

A model of human VPS34 CII was generated based on the *S. cerevisiae* homolog (PDB 5DFZ) (Rostislavleva *et al*, 2015) (Fig 3D). The 6 phosphoacceptors are all located away from the lipid and membrane binding regions of the complex (Herman *et al*, 1991; Panaretou *et al*, 1997; Rostislavleva *et al*, 2015; Chang *et al*, 2019) and are predicted to be surface facing. As 5 are on the same face of the complex (with S1039 on the opposing side), ULK-dependent multisite phosphorylation might significantly alter the electrostatic environment of this face of the complex. Favouring kinase access, S813, S861, S865 and S879, mapped to disordered regions. Although well conserved across multiple lineages, the majority are not conserved in *S. cerevisiae* (Rostislavleva *et al*, 2015).

### Deletion of VPS15 in a human cell model

To study the novel ULK-VPS15 signalling axis, CRISPR-Cas9 genome editing was used to ablate VPS15 (gene name *PIK3R4*) from HEK293A cells. CRIPSR guides (sgA and C) targeting the first coding exon of *PIK3R4* were used. Total loss of VPS15 expression was not observed, revealing the protein is required for viability in HEK293A;

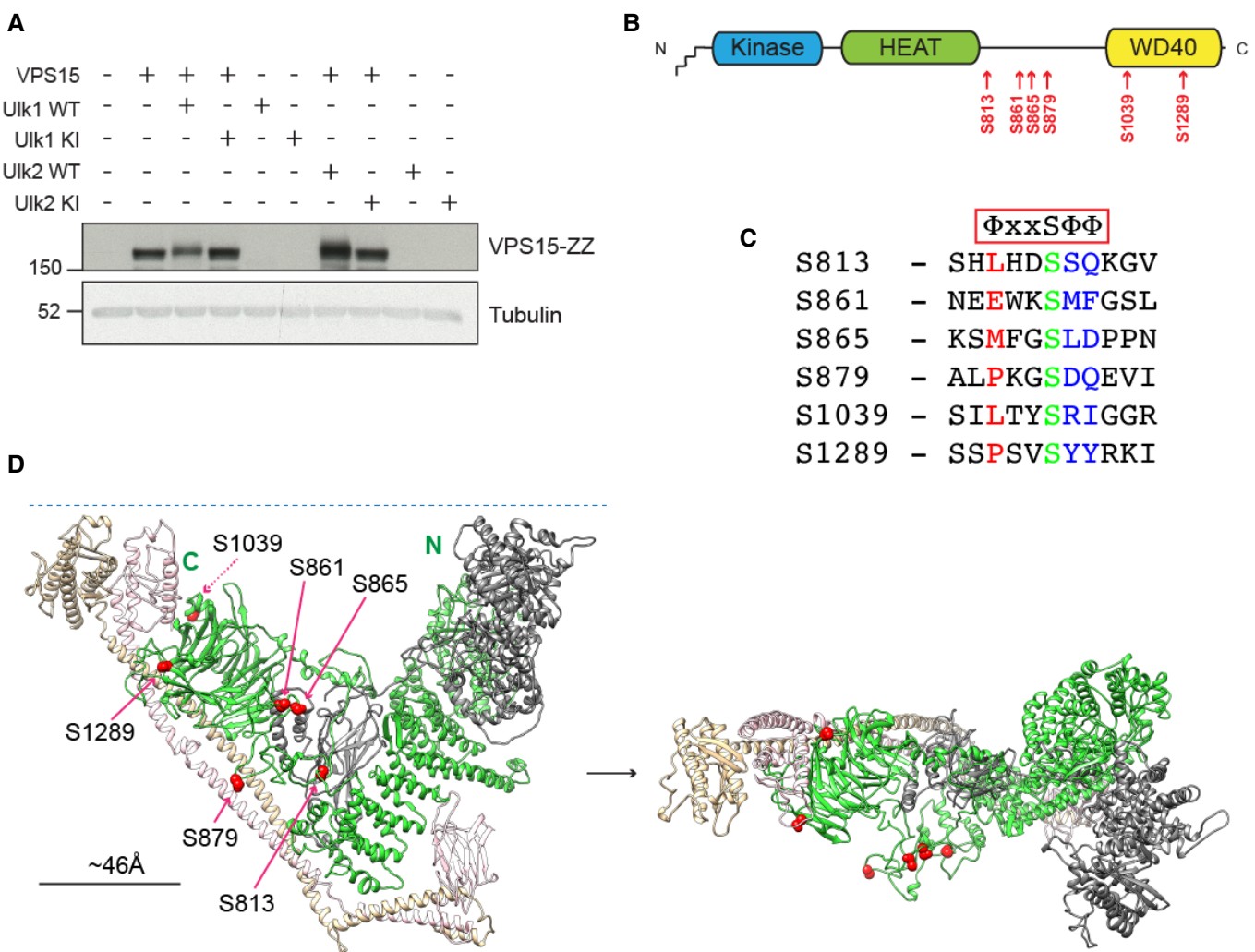

**Figure 3. VPS15 contains 6 putative ULK substrate residues.**

A  VPS15-ZZ was co-expressed with Ulk1 and Ulk2 wild type (WT) or kinase inactive (KI) before analysis by Western blot. An electrophoretic mobility shift in VPS15-ZZ was noted with WT only.

B  VPS15 is depicted schematically, with the N-terminal myristate and pseudokinase, HEAT and WD40 domains. The 6 potential ULK substrates are annotated below in red.

C  Alignment of human VPS15 sequences spanning each of the putative ULK substrates. Simplified rendering of the ULK consensus motif stipulating hydrophobics at positions −3, +1 and +2 is shown above (Φ = hydrophobic). These positions are highlighted in the sequence alignment (red −3, blue +1/+2) with the phosphoacceptor in green.

D  Human VPS34 complex II homology model showing VPS15 (green), VPS34 (grey), Beclin1 (beige) and UVRAG (pink) and potential ULK substrates (red spheres). (Left) Y-shaped VPS34 complex II in standard orientation with N- and C-termini of VPS15 labelled. The blue dashed line represents a lipid membrane, positioned close to the membrane binding (left- and right-hand tips of Y) and lipid kinase (right-hand tip of Y) domains of the complex. (Right) Rotated 90 degrees on horizontal axis towards point of view.

Source data are available online for this figure.

however, expression was largely abolished in 7 CRISPR clones (Fig 4A). VPS15 removal reduced the expression of ATG14, Beclin1 and VPS34, indicative of VPS34 complex destabilisation. All sgA-derived clones accumulated p62 and LC3 suggesting that autophagic flux was inhibited.

Whilst parental HEK293A and CRISPR control cells formed WIPI2 puncta after 1 h amino acid starvation, puncta formation was abolished in sgA-A8, sgA-B3, sgA-D6, sgA-D11 indicating VPS34 CI activity was depleted (Fig 4B). Large cytoplasmic vacuoles accumulated in all VPS15 CRISPR clones likely representing the

endolysosomal vacuoles that accumulate after treatment with VPS34 inhibitors or loss of Vps34 (Johnson *et al*, 2006; Jaber *et al*, 2012; Compton *et al*, 2016; Dyczynski *et al*, 2018). Note that validation of VPS15 loss by immunofluorescence using a previously published antibody was not possible as it showed a strong Golgi-localised signal in the control and CRISPR cell lines indicating that this staining is likely nonspecific (Stoetzel *et al*, 2016) (Fig EV3A). Abrogation of WIPI2 puncta formation correlated with a redistribution of PI3P-binding probe GFP-2xFYVE from a punctate to cytoplasmic localisation (Fig 4C) and, as observed in Vps15⁻ᐟ⁻ MEFs

(Nemazanyy *et al*, 2013), p62 and LC3 accumulated in large structures that were cleared by re-expression of VPS15 (Fig EV3B).

The chronic PI3P depletion phenotype observed in CRISPR clones sgA-A8, sgA-B3, sgA-D6 and sgA-D11 confirmed that VPS15 function was largely ablated in these clones, now referred to as VPS15 KOs. A genotypic analysis revealed that each clone had between 2 and 5 unique *PIK3R4* alleles detectable, at least 1 of which was unmodified (Fig EV3C). Intriguingly, VPS15 KOs had a majority of alleles in which valine 50 was deleted, with V50 lying in the predicted ATP-binding region of VPS15's pseudokinase domain (Fig EV3C and D).

To confirm loss of autophagic flux, VPS15 KOs expressing empty vector, VPS15 WT or VPS15 ΔV50 were starved for 1 h with 100 nM Bafilomycin A1 (Fig EV3E). Flux was blocked in VPS15 KOs and was rescued upon re-expression of VPS15. Notably, expression of VPS15 ΔV50 rescued neither autophagic flux nor VPS34 destabilisation in VPS15 knockouts.

### The ULK-VPS15 signalling axis regulates VPS34 activity

We then used the VPS15 KO model to assess the consequence of ULK-dependent VPS15 phosphorylation. KO cells were transiently transfected with VPS15-HA WT, 6SA or 6SE (to mimic stable dephosphorylation or phosphorylation of all ULK phosphoacceptors, respectively), with VPS15 6SA and 6SE expression resulting in a 20–30% reduction in starvation-induced WIPI2 puncta compared to WT-rescued cells (Fig 4D).

We examined the consequences of ULK phosphorylation at the molecular level. The rescue data suggested that ULK phosphorylation of VPS15 might modulate VPS34 lipid kinase activity. Because of its central role in autophagy initiation, we focussed on VPS34 CI. VPS15 KO cells were rescued with VPS15 WT, 6SA or 6SE before co-immunopurification of VPS34 CI for lipid kinase assays on ~800 nm large unilamellar vesicles (LUVs). Compared to WT, incorporation of both mutants reduced VPS34 CI lipid kinase activity (Fig EV4A), and VPS34 coimmunoprecipitation was slightly reduced with VPS15 6SA (Fig EV4B). To examine the role of phosphorylation more precisely, VPS15 WT, 6SA and 6SE were used to reconstitute VPS34 CI *in vitro*. Whilst incorporation of either mutant had little impact on CI stability in purified complexes (Fig 4E), lipid kinase activity

on giant unilamellar vesicles (GUVs) was greatly reduced (Fig 4F), although both kinase activity (Fig EV4C) and membrane binding (Fig EV4D) on 100 nm LUVs was unaffected.

### Serine 861 is the major ULK phosphoacceptor *in vitro*

To demonstrate direct phosphorylation of VPS15 by ULK, VPS34 CI components were overexpressed and immunopurified via ATG14 before *in vitro* phosphorylation by Ulk1. As full-length Ulk1 has a similar molecular weight to VPS15, a catalytically active fragment (Ulk1 1–427) was used. ATG14 was strongly phosphorylated as expected (Park *et al*, 2016), as well as a phosphoprotein at 150 kDa confirmed as VPS15 by Western blot (Fig 5A). VPS15 was phosphorylated *in vitro* with comparable efficiency when incorporated into VPS34 CII, with these data unexpectedly revealing UVRAG as an *in vitro* substrate of Ulk1 (Fig 5B). The ULK-dependent phosphorylation of the VPS15 phosphomutants was then compared. Phosphorylation was abolished for VPS15 6SA and 2SA (S861/865A) and comparison of the single phosphomutants identified S861 as the major *in vitro* phosphoacceptor (Fig 5C).

To confirm ULK-dependent phosphorylation in cells, phosphospecific antibodies were raised against VPS15 S861. VPS34 CI (Fig 5D) or CII (Fig 5E) components were co-expressed with Ulk1 1–427 WT or KI and cells were cultured for 1 h in the presence or absence of MRT68921 before lysis and complex coimmunoprecipitation via ATG14 (CI) or UVRAG (CII). A 4–8-fold increase in phosphorylation was observed in the presence of Ulk1 WT, which was not observed in cells treated with MRT68921 or expressing Ulk1 KI. Loading controls revealed that ATG14 and UVRAG underwent band shifts indicative of ULK-dependent phosphorylation in cells, corroborating data in Fig 5B to identify UVRAG as a novel ULK substrate (Fig 5F).

### VPS15 S861 phosphorylation reduces both autophagy initiation and autophagic flux

To understand how S861 phosphorylation could regulate autophagy, VPS15 KOs were stably transduced with constructs encoding WT, S861A, S861E or ΔV50 VPS15, or GFP as a control. Quantification of WIPI2 puncta after starvation revealed a significant reduction in

---

**Figure 4. VPS15 phosphorylation affects autophagy initiation and PI3P formation.**

A Lysates from 3 control clones and 7 VPS15 CRISPR clones were analysed by Western blot. LC3 and p62 accumulation as well as depletion of VPS34 CI components was observed in all 6 sgA-derived clones.

B Controls and 6 VPS15 CRISPR KO clones were starved for 1 h, fixed and imaged for WIPI2 (green). Depletion of VPS15 was associated with a block in WIPI2 puncta formation and large vacuoles (red asterisks). Quantification of WIPI2 spots, mean ± SEM, *n* = 3.

C GFP-2xFYVE was overexpressed in control (A3) and VPS15 KO clones. The GFP-2xFYVE puncta were abolished in VPS15 KO clones.

D Wild type (WT), 6SA or 6SE VPS15-HA, or empty vector (EV) were expressed in VPS15 KO or HEK293A cells. Cells were starved for 1 h and the distributions of WIPI2 (green), HA (red) and LC3 (blue) were assessed, mean ± SEM, *n* = 4.

E (Top) VPS34 CI incorporating VPS15 WT (black), 6SA (blue) or 6SE (pink) were purified, and thermal stability was assessed. The first derivative of the ratio of tryptophan emission at 330 and 350 nm (Y axis) is plotted against temperature (X axis, °C) with threshold melting temperatures annotated. (Bottom) Representative Coomassie-stained gel shows relative protein levels used.

F *In vitro* lipid kinase activities of WT-, 6SA- or 6SE-reconstituted VPS34 complex I on GUVs (giant unilamellar vesicles) were assessed. Representative images show GUV membranes (Lissamine Rhodamine-PE, green) and PI3P (Alexa Fluor 647-PX, red). Bottom left: VPS34 complex activities over time measured by recruitment of Alexa Fluor 647-PX to GUVs are shown. Bottom right: kinase reaction rates for GUVs phosphorylated with WT, 6SA or 6SE complexes, taken from linear portion of reactions. Representative data from 1 of 4 independent repeats are shown, with mean ± SD. In this experiment, 67 GUVs were analysed in total (15 from WT, 17 from 6SA and 35 from 6SE).

Data information: (B, D) **$P < 0.01$, ***$P < 0.001$ [one-tailed ANOVA]. (F) ***$P < 0.001$, n.s. not significant, [Student's *t*-test]. Scale bars 10 μm (B, D) and 5 μm (C, F). Source data are available online for this figure.

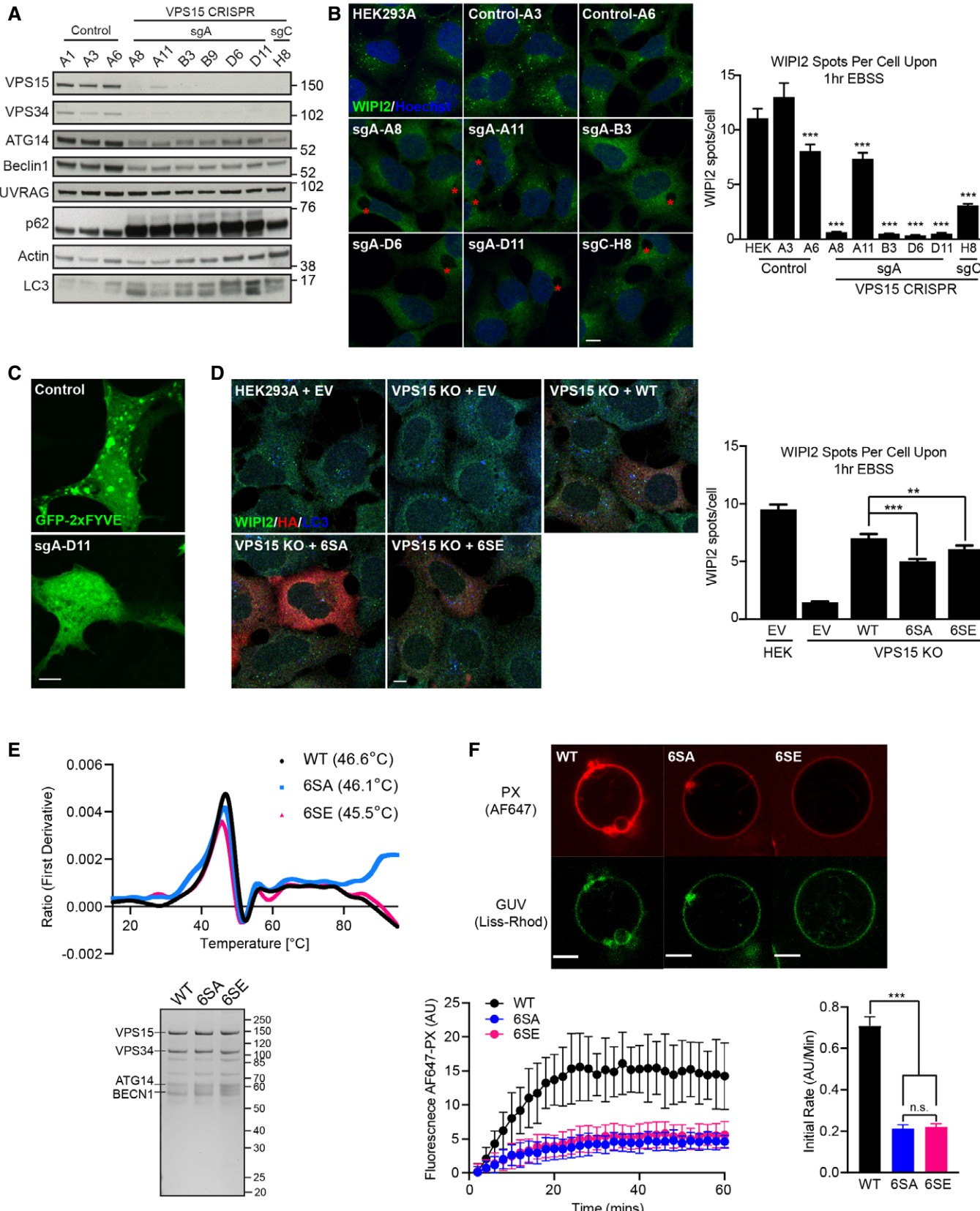

**Figure 4.**

VPS15 S861E expressing cells compared to WT (Fig 6A). To examine the degree to which serine 861 contributed to the total phosphomutant phenotype (Fig 4D), we transiently rescued VPS15 KOs with total phosphomutant VPS15 6SA/6SE or VPS15 5SA/5SE (note here S861 is not mutated). Whilst a 20–30% reduction in WIPI2 puncta formation was again observed in 6SA- and 6SE-expressing cells, no reduction in rescue efficiency compared to WT was noted in 5SA- or 5SE-expressing cells indicating that serine 861 mutations are the major driver of the total phosphomutant phenotype (Fig EV5A).

To test whether serine 861 phosphorylation status affects autophagic flux, the stably rescued VPS15 KO cells described in Fig 6A were starved for 1 h $\pm$ 100 nM Bafilomycin A1 followed by Western blot analysis. WT VPS15 rescued autophagic flux, whilst rescue efficiency was reduced in cells expressing S861A or S861E (Fig 6B), with the same result observed when a second VPS15 KO clone was stably rescued, indicating that the observation was not clone-specific (Fig EV5B). Finally, a time course experiment confirmed that rescue with S861E significantly inhibited flux after prolonged amino acid starvation (Fig 6C).

Taken together, these results suggest that ULK-dependent phosphorylation of VPS15 at serine 861 regulates starvation-induced autophagy, potentially by modulating the rate of PI3P production by VSP34 at omegasomes.

### Loss of VPS15 drives accumulation of distinct sets of ULK substrates

We then asked whether phosphorylation of the ULK substrate VPS34 S249, which strongly promotes binding to LC3/GABARAP proteins but for which no stimulus has been identified (Egan *et al*, 2015; Birgisdottir *et al*, 2019), was affected by VPS15 S861 phosphorylation status. No phosphorylation of VPS34 S249 was observed in both parental HEK293A and VPS15 KO cells rescued with VPS15 WT or S861 phosphomutants; however, it was enriched in VPS15 KO cells, made more apparent by reduced VPS34 levels (Figs 6C and EV5B). We decided to investigate this unexpected observation further and found that both amino acid starvation and treatment with the iron chelator deferiprone (DFP), which drives Parkin-

independent mitophagy (Allen *et al*, 2013), also promoted VPS34 phospho-S249 accumulation upon VPS15 removal (Figs 6C and EV5C). Cotreatment with DFP and MRT68921 led to a slight reduction in VPS34 S249 phosphorylation, suggesting that both ULK-dependent and ULK-independent phosphorylation may occur upon DFP-treatment (Fig EV5C). Importantly, we observed that iron chelation promoted VPS34 S249 phosphorylation even when VPS15 expression was intact. Phosphorylation was detected in parental HEK293A upon DFP treatment but not upon cotreatment with the Parkin-dependent mitophagy inducers Oligomycin and Antimycin (Fig EV5D). These data implicate phosphorylation of the ULK substrate VPS34 S249 as a biomarker for VPS15 depletion, in both iron and amino acid starvation responses and potentially in Parkin-independent mitophagy.

The unexpected accumulation of VPS34 phospho-S249 in VPS15 KOs led us to test whether this was common to multiple ULK substrates. We focussed on 2 examples, ATG13 S318 and PRKAB2 S39. Furthermore, we asked if accumulation of ULK phospho-substrates resulted from the chronic reduction in VPS34 activity. To test this, we performed a time course with VPS34-IN1 (IN1), a selective small molecule inhibitor of VPS34 (Bago *et al*, 2014). HEK293A cells were treated with IN1 up to 18 h, in the presence or absence of MRT68921 (Fig 7A). IN1 treatment did not promote VPS34 S249 phosphorylation, rather phosphorylation was only observed in VPS15 KO cells and in HEK293A transfected with VPS15-targeting siRNAs (Fig EV5E). However, both VPS15 ablation and IN1 treatment promoted accumulation of ATG13 phospho-S318 and PRKAB2 phospho-S39. In cells treated with IN1 for 18 h, phospho-substrate accumulation was substantially reduced by cotreatment with MRT68921, but was unaffected when ULK activity was inhibited for 1 h immediately pre-lysis (Fig 7A). These data reveal that at least two subsets of phosphorylated ULK substrates accumulate upon loss of VPS15, those that can (ATG13 phospho-S318 and PRKAB2 phospho-S39) or cannot (VPS34 phospho-S249) be recapitulated by inhibiting VPS34 activity. Notably, VPS15 ablation also led to an electrophoretic mobility shift, indicative of phosphorylation, in the ULK substrate ATG14 supporting a general increase in ULK kinase activity in VPS15 KOs (Fig 4A) (Park *et al*, 2016).

---

**Figure 5. Serine 861 is the major ULK phosphoacceptor in VPS15.**

A VPS34 CI was phosphorylated *in vitro* by Ulk1 1–427. Co-expressed VPS34, VPS15, Beclin1 and ATG14-ZZ were immunopurified and incubated with Ulk1 1–427 *in vitro*. The samples were then separated by SDS–PAGE and transferred onto PVDF membrane. *In vitro* phosphorylated proteins were visualised by autoradiography before VPS15 was visualised by immunoblotting, revealing that VPS15 and ATG14-ZZ were phosphorylated in this assay.

B All subunits for VPS34 CI containing ATG14-ZZ or CII containing UVRAG-ZZ were phosphorylated *in vitro* by WT or KI Ulk1 1–427 as indicated in A. Autoradiogram shows VPS15, ATG14-ZZ, UVRAG-ZZ and Ulk1 1–427 are phosphorylated. Phosphorylation occurred in reaction mixtures containing Ulk1 1–427 WT. The Coomassie-stained gel shows VPS15 levels in assay.

C WT VPS15 or VPS15 with 6SA, 2SA or individual phosphoacceptors mutated to alanine were incubated *in vitro* with Ulk1 1–427 and analysed as in B. The Coomassie-stained gel shows VPS15 levels in assay. Mean $\pm$ SEM, $n = 3$.

D VPS34 CI components expressed with Ulk1 1–427 WT or KI as indicated. After starvation for 1 h in the presence or absence of MRT68921 (ULK inhibitor), cells were lysed and VPS34 CI coimmunoprecipitated via ATG14-ZZ. VPS15 S861 phosphorylation was increased in the presence of active Ulk1. VPS15 pS861/total VPS15 was quantified, mean $\pm$ SEM, $n = 3$.

E HEK293A were transfected with VPS34 CII components and Ulk1 1–427 WT or KI and treated as indicated before VPS34 CII co-immunopurification via UVRAG-ZZ. Samples were analysed by Western blot as in D, revealing that Ulk1 can similarly phosphorylate VPS15 at S861 when incorporated into CII. Quantifications show mean of four independent experiments $\pm$ SEM, ***$P < 0.001$ [one-tailed ANOVA].

F Representative loading control samples from experiments in D and E were analysed by immunoblot. UVRAG and ATG14, each visualised using the affinity of the ZZ tag for 2° antibodies alone, underwent electrophoretic mobility shifts in the presence of active Ulk1 only, indicative of direct phosphorylation in cells.

Data information: (A, B) dashed lines indicate where autoradiograms were cropped for presentation. *$P < 0.05$, **$P < 0.01$, ***$P < 0.001$, n.s. not significant, [one-tailed ANOVA].
Source data are available online for this figure.

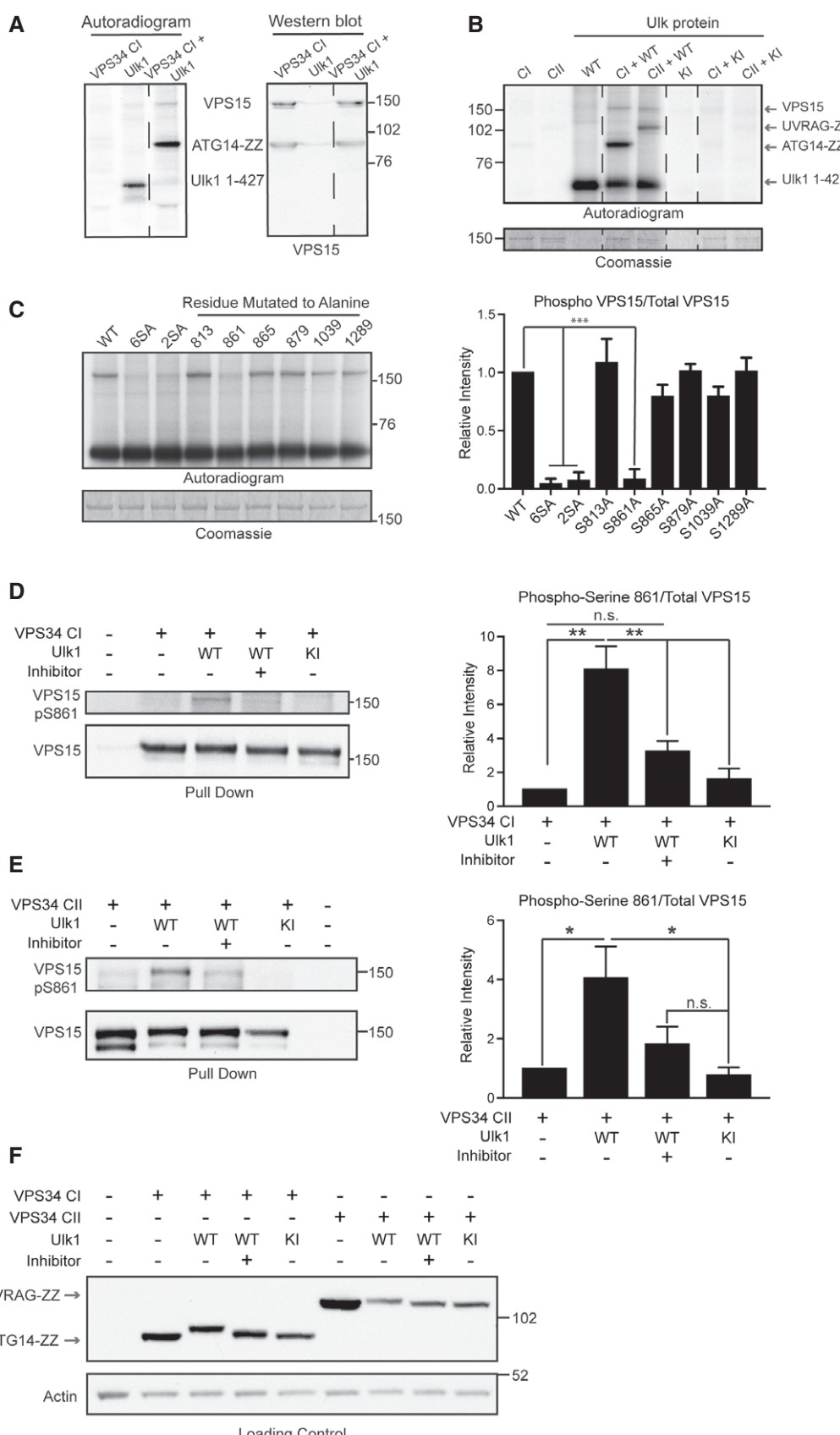

**Figure 5.**

### Early autophagy proteins accumulate in a VPS34 CI- and ULK-dependent manner in VPS15 KO cells

These findings caused us to re-evaluate the VPS15 KO phenotype. ULK substrate accumulation correlates with the formation of LC3-positive structures, which colocalise with p62 (Figs 6A and EV3B) and have been shown to be stalled autolysosomes (Nemazanyy *et al*, 2013) and protein aggregates (Lindmo *et al*, 2008; Jaber *et al*, 2012). We speculated that the accumulation of ULK phospho-substrates and LC3-/p62-positive structures in VPS15 KO cells were linked.

Previous studies have revealed that loss of key autophagy proteins causes accumulation of abnormal structures at autophagosomal formation sites on which early autophagic machinery coalesce (Kishi-Itakura *et al*, 2014). In FIP200 KO cells, p62 aggregates were shown to associate with peripheral ATG9A vesicles and in close proximity to ferritin clusters, and in ATG9A KO cells, exogenously expressed ULK1 localised around the periphery of such structures. Importantly, chronic (6 h) VPS34 inhibition with Wortmannin led to the accumulation of p62, ULK1, ATG9A and Ferritin and the loss of LC3-positive isolation membranes at initiation sites (Kishi-Itakura *et al*, 2014). More recently, direct association of FIP200 with p62 was shown to be crucial for aggrephagy of phase-separated p62 structures, which accumulate when autophagic turnover is blocked (Turco *et al*, 2019). Pertinently, the observation that Wortmannin treatment promoted FIP200-p62 colocalisation supports direct recruitment of early autophagic signalling complexes to p62-positive cargo to promote ULK-dependent autophagy (Turco *et al*, 2019).

We therefore hypothesised that the blockage of autophagic flux resulting from chronic reduction in PI3P promotes the sequestration of ULK to LC3-/p62-positive cargo, which could then act as hubs of ULK activity. Supporting this hypothesis, FIP200 localised to p62-positive structures (Fig 7B), which were also positive for Ferritin and Ubiquitin (Fig 7C). These structures also colocalised with the lysosome marker LAMP1 (Fig 7D). Whilst little colocalisation of ULK1 and ATG9A occurred in HEK293A cells, clear colocalisation in structures excluding GM130 was observed in VPS15 KOs (Fig 7E).

As MRT68921 cotreatment blocked the IN1-driven ULK substrate accumulation (Fig 7A), we investigated whether ULK activation status regulated its recruitment to the aberrant autophagic structures. 18 h IN1 treatment promoted formation of the structures in HEK293A and their enlargement in VPS15 KOs (Fig 8A). Super-resolution microscopy revealed that ULK1 and ATG9A were highly colocalised at the periphery of such structures, whereas p62 labelled the interior. Strikingly, cotreatment with MRT68921 increased the recruitment of both ULK1 and ATG9A, whilst driving a redistribution of both proteins from the periphery towards the interior (Fig 8A). These data reveal that ULK kinase activity status regulates the recruitment and distribution of autophagy proteins on aberrant structures that form upon chronic VPS34 inhibition or VPS15 ablation.

Finally, we depleted VPS34 CI- (ATG14) and CII- (UVRAG) specific components to test whether either complex was dispensable for either ULK substrate accumulation and/or recruitment of ULK complex components to the structures. Furthermore, directed by the findings of Turco *et al* (2019), p62 was depleted before IN1 treatment to test whether it was required for the novel phenotypes

(Figs 8B and C). ATG14 removal sensitised cells to IN1, with an increase in ATG13 phospho-S318 relative to controls noted both basally and after 18 h IN1 treatment. Furthermore, a greater than 2-fold increase in FIP200-positive structure formation upon IN1 treatment was observed after ATG14 depletion. IN1-induced ATG13 phospho-S318 accumulation was dampened in UVRAG depleted cells (in which CI formation was possibly favoured). Notably, p62 knockdown did not affect the accumulation of ATG13 phospho-S318 or FIP200-positive structure number (Figs 8B and C). Together, these data indicate that VPS34 CI stability negatively correlates with both ATG13 phospho-S318 level and accumulation of aberrant autophagic structures, with both phenotypes occurring in a p62-independent manner.

## Discussion

### Phosphoproteomics of ULK DKO cells reveals novel ULK substrates

We studied the ULK phosphoproteome assembled from complementary and unbiased SILAC and TMT screens for novel substrates to address the role of ULK in physiological and pathophysiological processes. Triaging by peptide array-based *in vitro* kinase assays led to the identification of substrates identified via in cell data using endogenous proteins and validated by direct phosphorylation *in vitro*. These included key regulators of autophagic lipid signalling (VPS15), energy homeostasis (PRKAG2, PRKAB2) and endosome-Golgi traffic (VPS26B).

Confirmatory peptide array-based *in vitro* kinase assays validated several novel ULK phosphoacceptors: ACTG1 (S33), ANXA2 (S127), CARS (S307), CTNND1 (S920), F11R (S287), LAP3 (S238), NHSL1 (S190), PCM1 (S110), PRKAB2 (S44), PRKAG2 (S124), TBC1D1 (S627), VILL (S233), VPS26B (S302 and/or S304) and VPS15 (S861), with phosphoacceptors in CHEK1, RALGPS2, SCEL, SORBS2 and VIM mapped to within fifteen residues. By comparing phosphorylation efficiency of a wide range of substrates, our data strongly indicate that Ulk1 and Ulk2 have virtually identical specificities *in vitro*. This finding supports the weight of evidence suggesting they share a very similar substrate repertoire (Chan *et al*, 2009; McAlpine *et al*, 2013; Egan *et al*, 2015; Park *et al*, 2016).

We show that the disease-relevant AMPK complex component PRKAG2 is phosphorylated at S124 by both ULK and AMPK; thus, identifying it as a novel biomarker sensitive to nutrient, energy and growth factor status and part of a complex signalling network. Additionally, we showed that UVRAG is directly phosphorylated by ULK1 both *in vitro* and in cells. Whilst the consequence of ULK-dependent UVRAG phosphorylation was not examined, it is notable that UVRAG phosphorylation has been reported to regulate both autophagosome-lysosome fusion and autophagic lysosome reformation (Kim *et al*, 2015; Munson *et al*, 2015), and that both ULK1 and UVRAG are implicated in control of ATG9A trafficking (Young *et al*, 2006; Orsi *et al*, 2012; He *et al*, 2013), autophagosome-lysosome fusion (Kim *et al*, 2015; Wang *et al*, 2018b) and ER-Golgi traffic (He *et al*, 2013; Joo *et al*, 2016; Gan *et al*, 2017). As all remaining VPS34 complex components are phosphorylated by ULK, the identification of VPS15 and UVRAG as substrates position ULK as the master regulator of the VPS34 complex.

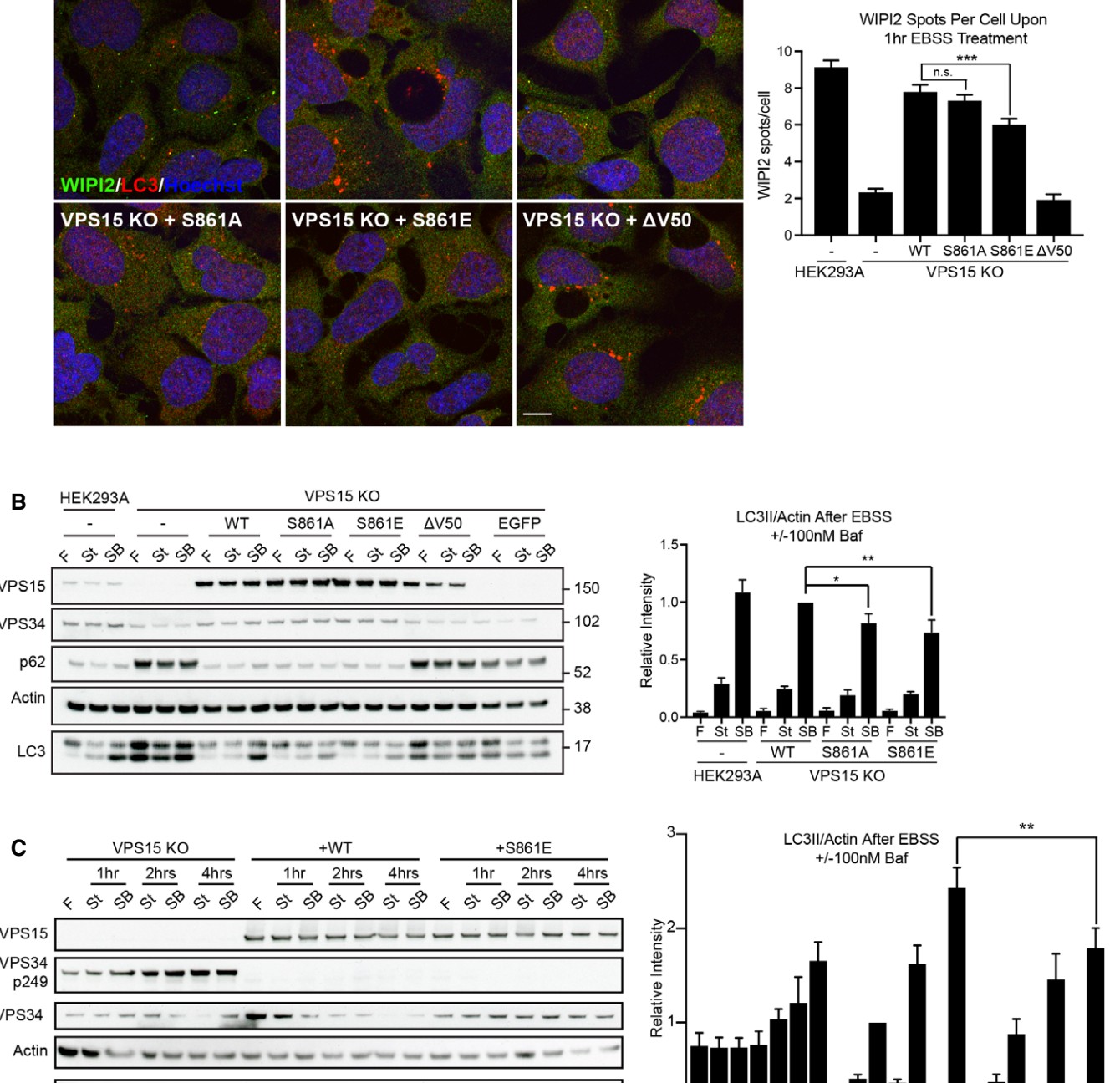

**Figure 6. ULK-dependent phosphorylation of serine 861-phosphorylation regulates autophagy.**

A   HEK293A, VPS15 KO control and stably rescued VPS15 KOs (sgA-B3) were starved for 1 h before WIPI2 (green), LC3 (red) and DNA (Hoechst, blue) were visualised.
    Quantification of WIPI2 puncta number per cell, mean ± SEM, *n* = 5.

B   Stably rescued VPS15 KO (sgA-B3), HEK293A and VPS15 KO control cells were starved for 1 h with (SB) or without (St) 100 nM Bafilomycin A1, or cultured in full media
    (F). Quantification shows LC3II/Actin, mean ± SEM, *n* = 5.

C   Autophagic flux examined over a 4-h starvation time course. LC3II/Actin was quantified, mean ± SEM, *n* = 6.

Data information: *P < 0.05, **P < 0.01, ***P < 0.001, [one-tailed ANOVA]. Scale bar 10 μm.
Source data are available online for this figure.

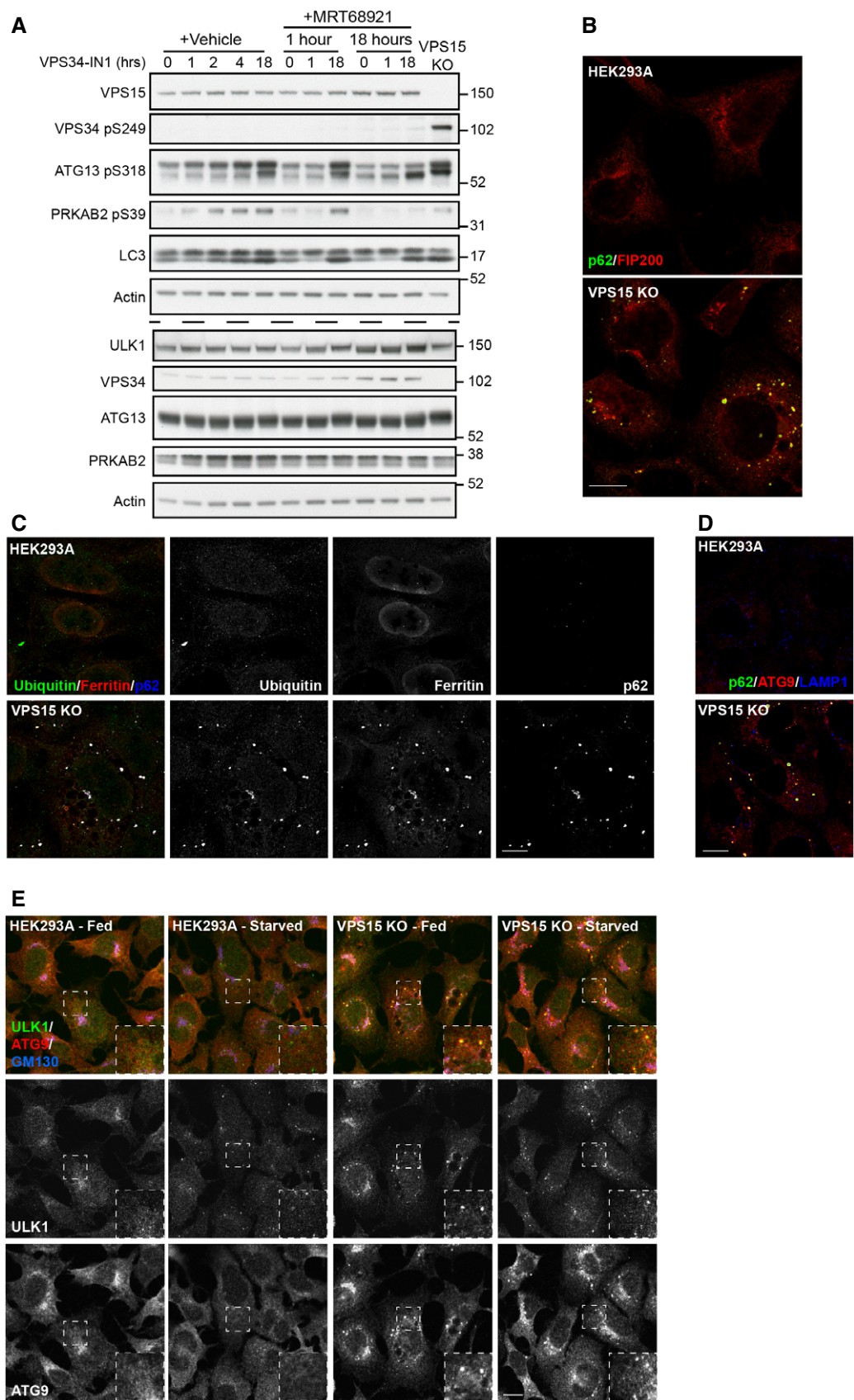

**Figure 7.**

**Figure 7.   VPS15 KOs accumulate phosphorylated ULK substrates and aberrant autophagic structures.**

A   HEK293A were treated with VPS34-IN1 (1 μM) alone or in the presence of MRT68921 (1 μM) for the indicated time. Untreated VPS15 KOs were included as a positive control and lysates analysed by Western blot. Identical samples were loaded in parallel for VPS34, ATG13, PRKAB2 analysis, indicated by dashed line.
B   HEK293A and VPS15 KOs were fixed and the labelled for p62 (green) and FIP200 (red).
C   HEK293A and VPS15 KOs were fixed and the labelled for ubiquitin (green), Ferritin (red) and p62 (blue).
D   HEK293A and VPS15 KOs were stained for p62 (green), ATG9 (red) and LAMP1 (blue).
E   HEK293A and VPS15 KO cells were fed, or starved for 2 h before fixation, and labelling with ULK1 (green), ATG9A (red) and GM130 (blue). Dashed boxes show magnified regions of interest.

Data information: Scale bars 10 μm.
Source data are available online for this figure.

Finally, we show that iron depletion, amino acid starvation and VPS15 stability are upstream stimuli for VPS34 S249 phosphorylation. To our knowledge, we are the first group to demonstrate VPS34 S249 phosphorylation without ULK1 overexpression.

## Regulation and phenotypic consequence of VPS15 phosphorylation by ULK

Within the ULK-VPS15 signalling axis, we identified 6 potential sites (S861 and S865 from SILAC and S813, S879, S1039 and S1289 in kinase-substrate overexpression experiments). S861 is the sole *in vitro* phosphoacceptor, which may reflect that ULK possesses a different sequence specificity *in vitro* or that the remaining sites are not kinase accessible after purification from cell lysates.

WIPI2 puncta formation and autophagic flux in cells and VPS34 kinase activity *in vitro* were reduced when WT VPS15 replaced with phosphomutant VPS15. The physiological stimulus governing phosphorylation was not identified, however, as activated ULK complex and VPS34 CI directly associate with omegasomes we choose to focus on the regulation of autophagy initiation. The validity of this choice is supported by the reduction in WIPI2 puncta observed after rescue with phosphomutant VPS15. However, it is possible that ULK1 targets multiple distinct VPS34 complex subpopulations, as has been reported for AMPK and mTORC1 (Kim *et al*, 2013; Yuan *et al*, 2013; Munson *et al*, 2015). Supporting this notion, we identified as an ULK substrate VPS26B, a component of the retromer complex which regulates endosome to Golgi transport in a VPS34 CII-dependent manner (Backer, 2016). As the retrieval of ATG9A from early endosomal and endolysosomal compartments was recently shown to occur in a VPS34- and retromer-dependent manner (Ravussin *et al*, 2021), potential roles of this novel ULK-VPS26B signalling axis are easily envisaged.

Our data indicate that the phosphorylation status of serine 861 is the major driver of the autophagy phenotypes observed in total phosphomutant-rescued VPS15 KO clones. We speculate that S861

mutational status was also responsible for the pronounced reduction in *in vitro* kinase activity when VPS15 mutant-containing VPS34 complexes were reconstituted. Rescue experiments generally produced similar phenotypes when serine-alanine and serine-glutamate VPS15 phosphomutants were compared. It is likely either that glutamate is unsuitable as a phosphomimetic in this instance or that cycles of phosphorylation and dephosphorylation are required to drive the relevant phenotype. We therefore conclude that ULK-dependent phosphorylation of VPS15, primarily at serine 861, promotes both VPS34 complex activity and autophagy.

Whilst we identified clear VPS15 phosphorylation-dependent phenotypes, the mechanism(s) of action remains elusive. Because 4 out of 6 phosphoacceptors lie in the unstructured HEAT-WD40 linker, predicted to allow targeting to membranes of varying curvature in yeast, we tested the effects of phosphomutant incorporation on kinase activity on liposomes of varying sizes. Incorporation of either VPS15 6SA or 6SE reduced VPS34 CI kinase activity on large LUVs and GUVs, which may mimic the outer surface of a growing phagophore. These data indicate that phosphorylation by ULK may control VPS34 complex flexibility and therefore substrate accessibility.

VPS34 activation requires conformational changes in VPS15 which culminate in the VPS34 lipid kinase domain dislodging from VPS15's pseudokinase domain. These require binding of dimerised NRBF2 at the base of the complex (Ohashi *et al*, 2016; Young *et al*, 2016; Young *et al*, 2019), which switches from an activator to an inhibitor of VPS34 CI activity upon mTOR phosphorylation, and is also an ULK binding partner and substrate (Behrends *et al*, 2010; Ma *et al*, 2017). These data indicate that together VPS15 and NRBF2 translate nutrient signals to the VPS34 lipid kinase domain to regulate its activity. No differences in NRBF2 binding to VPS15 6SA or 6SE were observed; however, it is possible that ULK phosphorylation of VPS15 may regulate productive docking of NRBF2 to VPS34 CI. Notably, NRBF2 binding leads to a conformational change in VPS15 between positions

**Figure 8.   Recruitment of autophagy proteins to aberrant structures regulated by both ULK complex and VPS34 complex I activity.**

A   HEK293A and VPS15 KOs were treated with vehicle or VPS34-IN1 (1 μM; IN1), alone or with MRT68921 (1 μM; MRT), for 18 h before fixation and visualisation of ULK1 (blue), p62 (green) and ATG9A (red). Dashed boxes show magnified regions of interest.
B   HEK293A were transfected with siRNAs targeting ATG14, UVRAG or p62, or with non-targeting siRNA (RISC-free, RF) and cells were treated with VPS34-IN1 (IN1; 1 μM) for 0, 4 or 18 h as indicated. Quantification of ATG13 pS318/Total ATG13, mean ± SEM, *n* = 3. Identical samples were loaded onto 2 separate SDS–PAGE gels for Western blot analysis, indicated by dashed line.
C   Cells transfected as in B were treated with IN1 for 18 h as indicated, fixed and labelled for ubiquitin (green), FIP200 (red) and p62 (blue). FIP200-positive bodies were quantified from four independent experiments with mean ± SEM plotted. Cells transfected with ATG14 siRNA generated significantly more FIP200 bodies per cell than all other conditions tested.

Data information: *P < 0.05, ***P < 0.001, [one-tailed ANOVA]. Scale bars 10 μm.

**A**

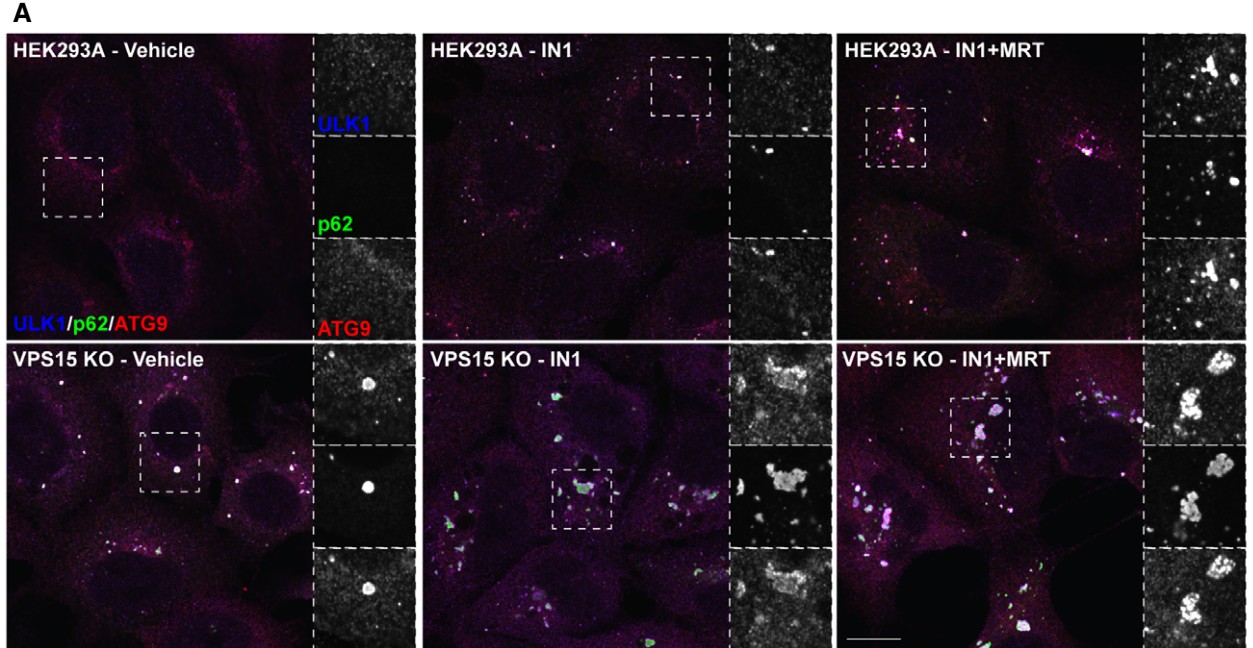

**B**

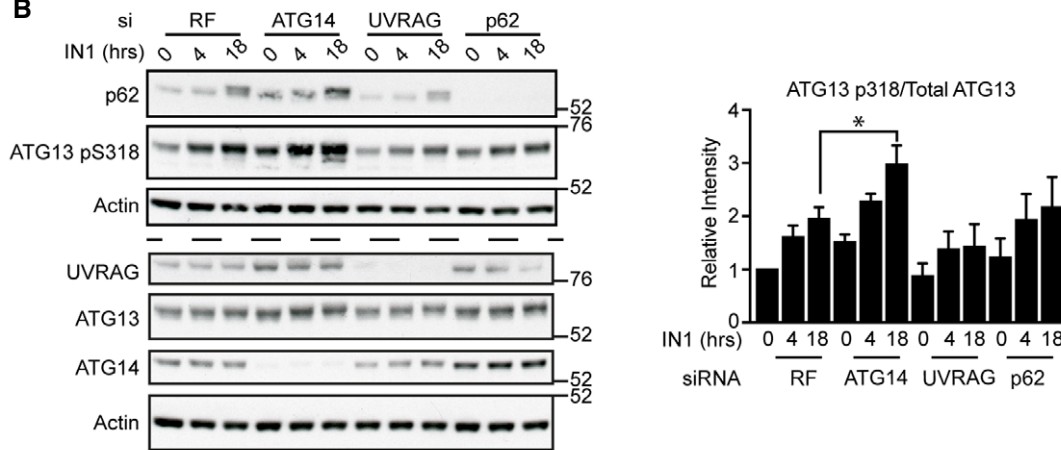

**C**

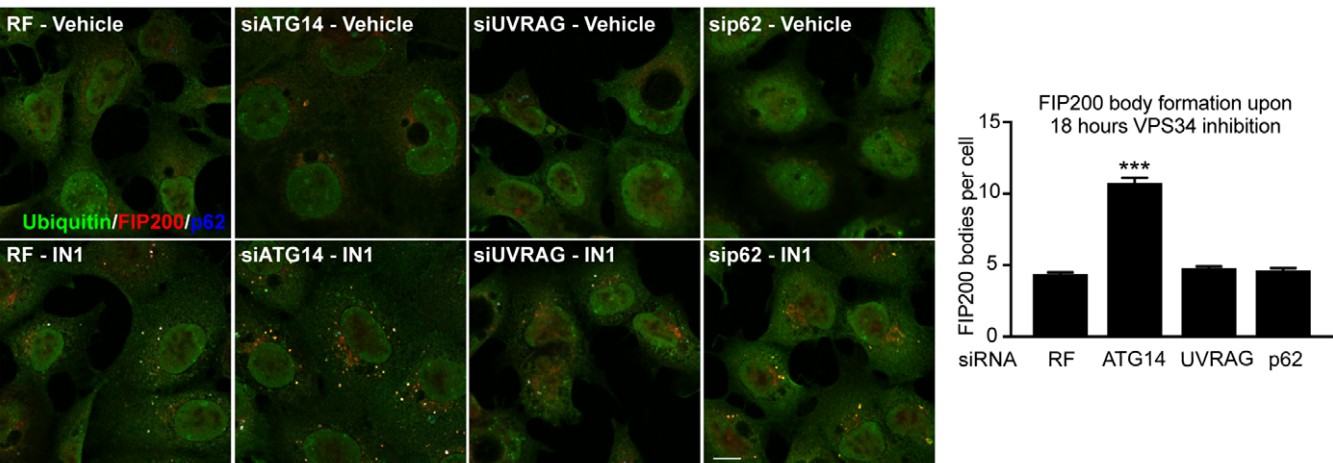

**Figure 8.**

864–882 (Young *et al*, 2016), which contains 2 ULK phosphoacceptors, and may be influenced by the phosphorylation of the neighbouring S861.

### VPS15 KOs reveals novel ULK-dependent phenotypes

VPS15 KO HEK293A cells largely phenocopy a previously published MEF line in which the pseudokinase domain was deleted. Nemazanyy *et al* (2013) concluded that the accumulation of p62-, LC3- and LAMP1/2-positive structures was due to inhibition of autolysosome clearance, a largely VPS34 CII-dependent process. Based on our examination of VPS15 KO and ATG14/UVRAG knockdown cells, we suggest these structures are similar to the aberrant autophagic structures reported by Kishi-Itakura *et al* (2014), and that significant reduction in VPS34 CI activity leads to accumulation and aggregation of autophagic cargo/adaptors, ubiquitin and unclosed autophagic membranes (Bjørkøy *et al*, 2005; Itakura & Mizushima, 2011; Sun *et al*, 2018; Zaffagnini *et al*, 2018), all but abolishing productive fusion of autophagosomes and lysosomes. By analysing our VPS15 KO model by super-resolution microscopy, we showed how ULK kinase inhibition resulted in a striking accumulation and redistribution of autophagy proteins on the structures. ULK inhibition promoted both recruitment of ATG9A, ULK1 and FIP200 to p62-/ubiquitin-positive bodies, and their intrusion towards the centre of the structures. We suggest that autophagy proteins localise to the surface of aggregates to promote their turnover by aggrephagy and that when turnover is blocked by ULK inhibition, their association time is increased promoting diffusion into the liquid-like p62-/ubiquitin-positive condensates (Sun *et al*, 2018; Zaffagnini *et al*, 2018; Turco *et al*, 2019).

Whilst it is poorly understood how ULK and VPS34 coordinate the spatial distribution of autophagy proteins at site of autophagosomal biogenesis, our data support a central role for ULK kinase activity. Unlike Karanasios et al, who showed that transient inhibition of VPS34 reduced ULK complex recruitment to such sites (Karanasios *et al*, 2013), we showed that chronic PI3P depletion enhances ULK recruitment p62-/ubiquitin-positive condensates, which are known to nucleate autophagosomes (Kageyama *et al*, 2021). These data therefore provide insights into both the ULK- and VPS34-dependent regulation of autophagy protein distribution on autophagic cargoes, and also on the biology of p62-/ubiquitin-positive condensates themselves, which were recently shown to coordinate antioxidant stress response via the sequestration of the NRF2-degrading E3 ubiquitin ligase KEAP1 (Kageyama *et al*, 2021).

The structures we report appear similar to disease-relevant and proteotoxic stress-induced aggregates which form in neurons described recently by Sarraf and colleagues, clearance which requires the autophagy adaptor TAX1BP1 (Sarraf *et al*, 2020). As we showed that p62 is dispensable for FIP200 recruitment to the aberrant autophagic structures, it is likely that alternative cargo adaptors such as TAX1BP1 are responsible for ULK complex recruitment.

Interestingly, we showed that the formation of these structures correlated with the starvation-independent accumulation of ULK phospho-substrates. We suggest that recruitment of ULK to the aberrant autophagic structures in basal conditions results in a high localised kinase activity via trans-autophosphorylation. This model is similar to that presented by Torggler *et al* (2016) who showed that

exogenous localisation of yeast Atg1p to a nonspecific scaffold as well as the clustering of cargo at the vacuolar membrane were both sufficient to drive local Atg1p activation.

As LC3B accumulated at the ubiquitin-positive structures, it is likely that GABARAP proteins were also present. As GABARAP and GABARAPL1 binding to ULK1 is known to promote its activation (Joachim *et al*, 2015; Grunwald *et al*, 2020), it is possible that they augment localised ULK activation. Phosphorylated substrates may be protected from phosphatases when localised to the structures, explaining why phosphorylation was not reversed after treatment with the ULK inhibitor MRT68921 for 1 h.

VPS34 phospho-S249 accumulation was not recapitulated upon chronic VPS34 inhibition but did occur upon VPS15 knockdown, which was especially striking given the reduction in VPS34 levels. As VPS34 alone is non-functional (Backer, 2016), it is possible that preventing association with VPS15 drives ULK-dependent phosphorylation of monomeric VPS34, promoting sequestration into autophagosomes via increased LC3/GABARAP binding (Birgisdottir *et al*, 2019) and thus maintaining the functional pool of VPS34.

### Conclusion

Our study identifies a wide range of novel and direct ULK substrates. We showed for the first time that VPS15 is regulated by phosphorylation, identified S861 as the major functional phosphoacceptor, and have shown that phosphomutant VPS15 reduces VPS34 activity *in vitro* and in cells. We have provided insight into the importance of the pseudokinase domain in human VPS15, which goes hand in hand with recent structural insights (Baskaran *et al*, 2014; Rostislavleva *et al*, 2015; Stjepanovic *et al*, 2017; Young *et al*, 2019). Whilst we do not yet understand how VPS15 phosphorylation affects VPS34 complex function, we propose that future work investigating alterations in protein interactors, VPS34 complex flexibility and/or NRBF2-dependent lipid kinase activation may provide further insight. Finally, we have identified 2 novel ULK-dependent phenotypes, namely the kinase activity-dependent recruitment and distribution of autophagy proteins on p62-/ubiquitin-positive structures and the accumulation of distinct subsets of ULK substrates upon removal of VPS15.

## Materials and Methods

### Cell culture

Buffers used for cell culture were Dulbecco's Modified Eagle Medium-High Glucose (DMEM, Sigma) and 0.25% trypsin-EDTA solution (Sigma), Earle's Balanced Salt Solution (EBSS) phosphate-buffered saline (PBS) and versene (0.02% (w/v) EDTA, 11 mg/l phenol red in PBS) (produced in house). Cell lines were cultured in DMEM supplemented with 10% foetal bovine serum (FBS, Sigma), 500 U/ml penicillin + 100 μg/ml streptomycin (Pen-Strep, Sigma) and 4.8 mM L-glutamine (full medium), and maintained in 10% $CO_2$. Short tandem repeat analysis was used to confirm the identity of every cell line utilised herein.

HEK293A, HEK293T and MEFs were described previously (Chan *et al*, 2007; Chan *et al*, 2009; McAlpine *et al*, 2013). Ulk1$^{-/-}$ Ulk2$^{-/-}$ (ULK DKO) MEFs used in the SILAC screen were described in

(McAlpine *et al*, 2013), and the DKO MEF line used in the TMT screen, VPS15 KO clones and lentivirally rescued VPS15 KO clones were generated as described below. CRISPR/Cas9-mediated genome engineering was used to generate the VPS15 KO HEK293A clonal cell lines following the Zhang laboratory protocol (Ran *et al*, 2013). The first coding exon of VPS15 (gene name *PIK3R4*) was targeted with 2 sgRNAs (sgA, forward 5′-CAA AAA CCT TCA CAA CGA CC-3′; reverse 5′-GGT CGT TGT GAA GGT TTT TG-3′; and sgC, forward 5′-ACA AAT CTG GAG TTC GTC AT-3′; reverse 5′-ATG ACG AAC TCC AGA TTT GT-3′), designed using the Zhang laboratory CRISPR design tool (now offline). sgRNAs were cloned into Bbs1 site of pSpCas9(BB)-2A-GFP plasmid also obtained from the Zhang laboratory (Addgene; 48138). Following single-cell sorting on GFP-positive cells, Western blotting was used to screen colonies for VPS15 deficiency. VPS15 KO clones (sgA-B3, sgA-D6 and sgA-D11) were used interchangeably between experiments.

Treatments were applied for specified times, diluted in either EBSS or full medium: Torin 1 (100 nM, Cayman Chemical), MRT68921 (1 μM, Sigma), 991 (1 μM, Selleckchem), Oligomycin (1 μM, Sigma), Deferiprone (1 mM, Sigma), Antimycin (1 μM, Sigma), Bafilomycin A1 (100 nM, Calbiochem) and VPS34-IN1 (1 μM, Cayman Chemical).

### Transfections/Transductions

For all experiments other than VPS15 KO transient rescue experiments, in which JetPRIME (Polyplus) was used, Lipofectamine 2000 (Life Technologies) was used for plasmid transfections, with both reagents used as per the manufacturer's instructions. For knockdown experiments, 20 μM siRNA stocks were made up in siRNA dilution buffer (Dharmacon) before transfection at 50 nM final concentration using Lipofectamine 2000 (Life Technologies). Cells were reverse transfected with siRNA on day 1 and again on day 2 before preparation for analysis by Western blot or immunofluorescence on day 4.

Primary ULK DKO MEFs derived from day 13 embryos were cultured in full media supplemented with 20% FBS before immortalisation via retroviral transduction of SV40 large T antigen. When stably rescuing VPS15 KO clones, lentiviral transfer vectors (3[rd] generation) encoding wild type, S861A, S861E, VPS15-HA_IRES_EGFP or GFP alone were used. Pen-Strep was excluded from culture media and was replenished only after successful transduction. Retro- and lentiviruses were produced in HEK293T. Alongside lentiviral transfer plasmids listed above, psPAX2 gag-pol and VSVG plasmids were transfected at a 4:3:1 ratio. To generate retroviruses pBABE SV40T Puro, MMLVgag-pol and VSVG plasmids were transfected at a 3:2:1 ratio. Full medium was replenished after transfection, and after 24 h, virus-containing media were collected and supplemented with Polybrene (8 μg/ml, Sigma). The virus-containing media were centrifuged (4,000 *g*, 5 min) and filtered (0.45 μm Millex HV filter (Millipore)) before addition to target cells. If lentiviral transduction efficiency was poor, this process was repeated ≤ 3 times, with virus-producing HEK293T discarded at 3 days post-transfection. Fluorescence-activated cell sorting (FACS) was used to isolate the lowest EGFP expressing lentivirus-transfected cells 5–7 days after transduction. Puromycin (1 μg/ml, Sigma) was added to growth media of retrovirally immortalised cells for 7 days to isolate successfully transduced cells.

### Constructs and siRNAs

Constructs encoding VPS15-ZZ, VPS34, ATG14+Beclin1, VPS34 complex I (multi-cassette complex encoding VPS34, VPS15, Beclin1 and ATG14-ZZ), VPS34 complex II (multi-cassette complex encoding VPS34, VPS15, Beclin1 and UVRAG-ZZ) and GST-PX were kindly provided by Roger Williams PhD. Phosphomutant and HA-tagged VPS15 constructs were generated in this study. The expression plasmid for UVRAG was provided by Christian Behrends PhD and the myc tag added by Harold Jefferies PhD. Myc-ULK1 1–427 kinase inactive was generated in this study, and all other myc-ULK constructs were reported previously (Chan *et al*, 2007; Chan *et al*, 2009). FLAG-PRKAG2 was from Grahame Hardie PhD, whilst FLAG-PRKAG2 Δ124 and all PRKAG2-GFP constructs were generated in this study. PRKAA1-myc and PRKAB2 were kindly provided by David Carling, PhD. GFP-2xFYVE was a gift of George Banting PhD was from Roger Williams PhD. The lentiviral transfer vector encoding WT VPS15 was purchased from VectorBuilder, with all variants generated in this study. RISC-free siRNAs (D-001220-01) or siGENOME SMARTpools targeting ATG14 (M-020438-01), UVRAG (M-015465-01) or p62 (M-010230-00) were ordered from Dharmacon.

### Antibodies

Anti-LC3-B (ab48394) and beta tubulin (ab6046) were from Abcam, and anti-actin AC40 (A4700) was from Sigma-Aldrich. Anti-ATG14 (M184-3) was from MBL Life Science, and anti-VPS15 (NBP130463) was from Novus Biologicals. Anti-GM130 (610822), LAMP1 (CD107a), p62 (for use in Western blotting, 610833) and TIM23 (611223) were from BD Biosciences, and anti-p62 (for use in immunofluorescence, GP62-C) was from Progen Biotechnik. Anti-ULK1 (sc-33182) was from Santa Cruz Biotech. Anti-Beclin1 (3738), Phospho-PRKAA1 T172, VPS34 (3811), Phospho-VPS34 Serine249 (13857), PRKAB1/2 (4150) and Phospho-PRKAB2 Ser39 (82791) were from Cell Signalling Technology. Anti-Phospho-ATG13 Ser318 (600-401-C49) was from Rockland. HRP-conjugated anti-GST (RPN1236) was from GE Healthcare, and anti-Ferritin (ABIN99122) was from Antibodies-Online. Anti-FIP200 (17250-1-AP) was from Protein Tech. Anti-Phospho-VPS15 Serine 861 was generated during this study, as was Phospho-PRKAG2 Serine 124. Anti-WIPI2 mouse monoclonal (Polson *et al*, 2010), anti-mATG9 Armenian hamster monoclonal (Webber & Tooze, 2010) and anti-ATG13 (Chan *et al*, 2009) were previously described and both mouse monoclonal myc (9E10), and GFP (3E1) was from the Francis Crick Institute.

Secondary antibodies for immunofluorescence (anti-rabbit IgG Alexa Fluor 488, 555, and 647; anti-mouse IgG Alexa Fluor 488, 555, and 647) were from Life Technologies, and anti-hamster Cy3 was from Jackson ImmunoResearch. For use in Western blotting, HRP-conjugated secondary antibodies were from GE Healthcare.

### Generation of lysates for Western blotting

PBS was used to wash cells twice before lysis in TNTE buffer (20 mM Tris pH6.8, 150 mM NaCl, 5 mM EDTA, 1% w/v Triton X-100) containing 1X EDTA-Free Complete Protease Inhibitor cocktail (Roche) and 1× PhosSTOP (Roche). To prevent cell detachment, HEK293A that were treated with EBSS were not washed. Lysates were cleared by centrifugation at 13,200 *g* for 5 min. Post-nuclear

supernatants were mixed with 5× sample buffer (213.5 mM Tris-HCl pH 6.8, 50% w/v glycerol, 16% β-mercaptoethanol, 15% w/v SDS, bromophenol blue) to a final concentration of 1× before incubation at 100°C for 5 min. Lysates were then analysed by SDS–PAGE as described previously (Judith *et al*, 2019).

## Immunoprecipitation

For immunoprecipitation experiments, 10 cm dishes were washed twice with cold PBS before harvesting in 500 µl TNTE. Post-nuclear supernatants (PNS) were prepared as described above. A sample of PNS was retained and mixed with sample buffer comprising the loading control/input sample. The antibody-conjugated resins used are as follows: myc-Trap_A (ChromoTek), GFP-Trap A (ChromoTek), anti-FLAG M2 Affinity Agarose Gel (Sigma) and IgG Sepharose 6 Fast Flow affinity resin (GE Healthcare). To immunoprecipitate ULK complexes for peptide array-based *in vitro* kinase assays, protein G Sepharose beads (Sigma) were bound with anti-myc 9E10 by incubating 50 µl protein G Sepharose slurry per 10 cm dish with 15 µg antibody at 4°C, turning end over end for 60 min. All beads were washed twice in PBS and twice in TNTE before use.

Immunoprecipitation experiments were performed by rotating clarified lysates with washed beads at 4°C end over end for 2–4 h, after which beads were pelleted by centrifuging the lysate-resin mixture at 7,000 *g*. The supernatant (unbound sample) was retained to test pulldown efficiency. The pelleted beads were washed in TNTE containing 0.1% Triton X-100 (3–5 × 1 ml). If no further processing was required, the beads were aspirated and prepared for Western blot analysis.

To improve sample purity when purifying ULK complexes and PRKAG2, the wash buffer NaCl concentration was increased from 150 to 300 mM. Where expression levels of target proteins differed, immunoprecipitates sometimes had to be normalised prior to analysis. In these cases, plasmid amounts used for transfection were altered empirically until expression levels were within the same range. Where required, pcDNA3.1 (+) empty vector was added proportionately such that the total amount of DNA used for transfection was maintained.

## Protein elution for peptide array-based *in vitro* kinase assays

To generate kinase samples for use in peptide array-based *in vitro* kinase assays, HEK293A cells were cotransfected with ATG13-FLAG, FLAG-FIP200 and myc-Ulk1 or myc-Ulk2 in 10 cm dish format. Transfected cells were starved for 1 h ± MRT68921 as indicated and complexes were immunoprecipitated as described above. To elute ULK complexes, 100 µl kinase reaction buffer (KRB – 20 mM HEPES pH 7.4, 20 mM MgCl$_2$, 25 mM beta-glycerophosphate, 2 mM dithiothreitol, 100 µM sodium orthovanadate) per 10 cm dish supplemented with 0.5 mg/ml myc peptide (EQKLISEEDL) was added to fully aspirated beads before shaking for 20 min at 25°C. Equal protein amounts were confirmed by immunoblot. The elution buffer was supplemented with MRT68921 where appropriate, to a final concentration in the *in vitro* kinase assay reaction mixture of 1 µM.

## *In vitro* kinase assays

HEK293A expressing kinase (myc-Ulk1 WT/KI) or substrates (VPS34 complex I components, VPS15-ZZ alone or FLAG-PRKAG2

fragments) were lysed and proteins immunoprecipitated as described above. Kinase-expressing dishes were starved for 3,060 min before lysis. Washed beads were equilibrated in KRB before kinase- and substrate-bound beads were mixed or left unmixed as indicated. The reaction mixture (Radioactive Assays: 32 µl KRB with 100 µM ATP and 2 µCi ATP[γ-$^{32}$P], (Easy Tide Lead, Perkin Elmer); Non-Radioactive Assays: 32 µl KRB with 1.8 mM ATP]) was added to fully aspirated beads, before incubating for 30 min at 30°C. Sample buffer was used to terminate reactions, and samples were electrophoresed by SDS–PAGE. For radioactive assays, gels were fixed using Instant Blue Coomassie staining (Expedeon) as per manufacturer's instructions and gels were dried before using to expose autoradiography/chemiluminescence film (Amersham Hyperfilm ECL, GE). For non-radioactive assays, the reaction mixtures were analysed by immunoblot.

Eluted ULK complexes were used to phosphorylate peptides in array format (see above and Fig EV2). For each experiment, 2 identical array membranes (produced in house) were activated via 2 min incubation in methanol, rinsed well in wash buffer (20 mM HEPES pH 7.4, 0.02% Triton X-100) before 4 × 10 min incubations with wash buffer rocking at room temperature. Membranes were rocked overnight in blocking buffer (20 mM HEPES pH 7.4, 0.02% Triton X-100, 0.2 mg/ml BSA) at 4°C. The following day, reaction mixtures were generated by diluting kinase eluates (see above) to 5 ml with protein dilution buffer (0.05 mg/ml BSA in KRB) before mixing 1:1 with ATP solution (200 µM ATP, 200 µCi ATP[γ-$^{32}$P] in KRB) and adding to arrays. The peptide array membranes were rocked gently for 15 min at room temperature in 10 ml reaction mixtures before rinsing thoroughly in PBS. Membranes were then washed extensively by rocking at room temperature for 10 min in PBS ×2, 30% Glacial Acetic Acid (Fisher) ×3, dH$_2$O ×2, 0.1N NaOH ×1, dH$_2$O ×2, 30% Glacial Acetic Acid ×2. Phosphorylated peptide arrays were visualised using autoradiography/ chemiluminescence film (Amersham Hyperfilm ECL, GE).

## Phosphoantibody generation

15mer peptides centred on VPS15 serine 861 encoding a phosphoserine in the central position (NVNEEWK(pS)MFGSLDC) were generated in house and used to immunise rabbits. The antibodies were affinity-purified with the phospho-S861 peptide. The affinity-purified phosphoantibody was used in the presence of the non-phosphorylated peptide (NVNEEWKSMFGSLDC) at 5× molar ratio. The PRKAG2 phosphoserine 124 were were produced and purified by Covalab, FR.

## Confocal microscopy

For immunofluorescence analysis, cells were seeded on coverslips before fixation in 3% PFA. 5-min incubations in either methanol or 0.2% Triton X-100 in PBS were used to permeabilise cells, which were then blocked in 5% BSA in PBS. Cells were incubated for 1 h in in 5% BSA with the addition of appropriate primary and secondary antibodies, before mounting on slides and analysis using either a Zeiss Upright 710 confocal microscope with a 63× objective lens or, for data shown in Figs 7 and 8, a Zeiss Invert 880 confocal microscope with Airyscan.

## High throughput screening image acquisition and analysis

For the HTS experiments, cells were grown in 96-well plates (Greiner Bio-One Ltd 655090) before fixation, permeabilisation and antibody staining as described above. Imagining was performed using an Opera Phoenix High Content Screening System (PerkinElmer) with 40×/NA 1.1 water-immersion lens. Z-stacks from 0 to 1 μm with a step size of 0.5 μm were acquired using excitation lasers at 405, 488 and 555 nm, and emission filters at 450, 525 and 580 nm, respectively. Cell segmentation and quantification analysis were performed using Harmony software 4.9 detecting the parameters indicated in Appendix Table S3.

## Mass spectrometry

WT and ULK DKO MEFs were grown in DMEM supplemented with 10% dialysed FCS (Invitrogen) supplemented with a combination of either 100 mg per litre of light ($^{14}$N,$^{12}$C) or heavy ($^{15}$N,$^{13}$C) lysine and arginine (CK Isotopes), with full label incorporation confirmed by mass spectrometry before use. Cells were treated with EBSS or with SILAC media supplemented with Torin 1 (100 nM) for 2 h to induce ULK activation. Cells were detached and mixed in a 1:1 ratio heavy:light, and 5 mg total protein per mixture was used per experiment. Proteins were digested using a combination of trypsin and LysC and peptides were fractionated using Strong Cation Exchange chromatography using a Poly LC PolySULFOETHYL A 100 × 4.6 mm 5 μm 200A column. The flow rate was 1 ml/min and the gradients used volatile buffers A: 10 mM Ammonium Formate, 25% ACN, pH 3.0 and B: 500 mM Ammonium Formate, 25% ACN, pH 6.8. All fractions collected were taken to dryness by vacuum centrifugation. 5 mg of Titansphere titanium dioxide beads were used per SCX fraction for phosphopeptide enrichment as described previously (Swaffer *et al*, 2016). Dried phosphopeptides from the titanium dioxide enrichment were cleaned up in preparation for LC-MS/MS analysis using C18 Stage Tips and taken to dryness by vacuum centrifugation. Each sample was resuspended in 35 μl 1% TFA, sonicated for 15 min and injected three times (10 μl per injection). Peptide mixtures were separated on a 50 cm, 75 μm I.D. Pepmap column over a 3 h gradient and eluted directly into an LTQ-Orbitrap Velos mass spectrometer. The instrument ran in data-dependent acquisition mode with the top 10 most abundant peptides selected for MS/MS by either CID, MSA or HCD fragmentation techniques (one fragmentation technique per injection). MaxQuant and Perseus were used for data processing with false discovery rate (FDR) of 1% set at the protein, peptide and phosphosite level.

To generate samples for TMT analysis, SV40 immortalised WT and ULK DKO MEFs were starved for various time points as indicated before trypsinisation, washing in 0.1% (w/v) BSA PBS and pelleting via centrifugation (3,000 *g*, 1 min). Cell pellets were resuspended in ice-cold PBS, then pelleted again for lysis in 200 μl 8 M Urea lysis buffer (8 M Urea, 50 mM HEPES pH 8.2, 75 mM NaCl, 1X PhosStop (Roche), 1× Complete EDTA-Free Protease Inhibitor). Lysates were sonicated, and protein concentration was measured before 220 μg aliquots of each sample were digested in a combination of trypsin and LysC. Next, individual digests were cleaned and concentrated using Nest Group C$_{18}$ MacroSpin columns. Peptide labelling with TMT reagent was performed according to the manufacturer's protocol (TMT 10plex™, #90110, Thermo). For peptide enrichment, the Thermo Scientific High-Select Fe-NTA phosphopeptide enrichment kit (A32992) was used as described here (https://star-protocols.cell.com/protocols/66). Phosphopeptides eluted from the procedure were dried by vacuum centrifugation and subjected to fractionation using the High pH Reversed Phase Fractionation kit (Pierce, 84868). The resulting dried TMT labelled phosphopeptides were resuspended in 1% TFA and separated on a 50 cm, 75 μm I.D. Pepmap column over a 2–4 h gradient at 40°C and eluted directly into the mass spectrometer (Orbitrap Fusion Lumos). The instrument was run in data-dependent acquisition mode with the most abundant peptides selected for MS/MS by HCD fragmentation. MaxQuant v1.6.x was used to process the raw data acquired with a reporter ion quantification method. A protein, peptide and phosphosite estimated false discovery rate (FDR) of 1% was used with the *mus musculus* Uniprot KB database used for database searching. Reporter ion intensities were imported into Perseus from the Phospho STY Sites table for further downstream processing.

To measure changes in VPS15 phosphorylation upon Ulk1 overexpression, HEK293A were cotransfected with the multi-cassette VPS34 complex I construct (see above) alone, with myc-Ulk1 WT or with myc-Ulk1 KI. After lysing cells, IgG Sepharose 6 Fast Flow affinity resin was used to coimmuniprecipitate VPS34 complex I via ATG14-ZZ. Samples were electrophoresed on a 4–12% mini gel (NuPAGE) and gels were fixed (Instant Blue, Expedeon) before VPS15 bands were excised and subjected to overnight trypsin digestion. Dried peptides were resuspended in 0.1% fluoroacetic acid and separated on a 50 cm, 75 μm I.D. EasySpray C18 column over a 30-min gradient and eluted directly onto a Q-Exactive mass spectrometer. The instrument was run in data-dependent acquisition mode. MaxQuant and Perseus were used to process the data. Phosphorylation of serine, threonine and tyrosine and oxidation of methionine were used as variable modifications in the search settings. Potential ULK substrates were highly enriched in myc-Ulk1 WT-expressing cells only.

## Phosphoproteomics analysis

Analytical approaches for SILAC screen are described in the text. Bioinformatics analyses for TMT were performed as follows: three variables were defined to describe both the starvation and the ULK dependence of the phosphorylation events as shown in Fig EV1F. The comparisons selected were WT 0 min–WT 60 min (increase in phosphorylation after 60 min starvation in WT MEFs), WT 60 min–DKO 60mins (decrease in phosphorylation in DKO compared to WT MEFs at 60 min starvation) and WT 60 min–WT 60 min Refed (decrease in phosphorylation when full media was replenished in WT MEFs). Threshold values for each variable were established by trial and error, and phosphopeptides simultaneously satisfying all three were brought forward preferentially.

A fourth variable was used to annotate the data set, statistical similarity to Prkab2 S38. As ULK substrates may display varied phosphorylation dynamics, peptides with a correlation coefficient of > 0.7 were brought forward to maximise substrate identification (Pearson's correlation coefficients of > 0.8 are considered significant). Phosphopeptides satisfying the first three variables were given precedence when shortlisting putative ULK substrates as this was more stringent. However, as many of the sites shortlisted using

the four variables did not display profiles indicative of direct ULK phosphorylation the data set was curated manually with clear false positives removed.

### Protein purification

GST-PX for use in the GST overlay assays were purified from *E. coli* as described in (Wirth *et al*, 2019). Purification protocols of human complexes I and II were described previously (Ohashi *et al*, 2016; Ohashi *et al*, 2019). The derivative mutants were purified essentially in the same way as the WT complexes.

### Thermostability assays

Purified VPS34 complex I (WT or mutant) were diluted to 0.5 mg/ml in dilution buffer (20 mM HEPES 8.0, 150 mM NaCl, and 0.5 mM TCEP for complex I). For each sample, 10 μl of the diluted sample was absorbed in a glass capillary, then set on a holder for a differential scanning fluorimetry reader (Prometheus, NanoTemper). The thermal stability was measured in a range from 15 to 95°C following the manufacturer's instruction.

### GUV *in vitro* kinase assays

The lipid composition for the GUVs *in vitro* kinase assays was 18% liver PI, 10% DOPS, 17% DOPE, 55% DOPC, 0.03% DSPE-PEG-Biotinyl, 0.017% Liss-Rhodamine. GUV generation, activity measurement and image analysis were described previously (Ohashi *et al*, 2020).

### LUV *in vitro* kinase assays

The lipid composition for LUV *in vitro* kinase assays was 18% liver PI, 10% DOPS, 17% DOPE, 55% DOPC. For Fig EV4A, 800 nm LUVs generated via extrusion were incubated in lipid kinase reaction buffer (described previously; Ohashi *et al*, 2020) containing 200 μM ATP with VPS34 complex I incorporating WT, 6SA or 6SE VPS15-ZZ, coimmunoprecipitated via VPS15-ZZ. Reactions were shaken at room temperature for up to 30 min before spotting on PVDF. Membranes were blocked in PBST 3% BSA and PI3P visualised using the GST-PX probe (2 μg/ml, PBST 3% BSA). Bound GST-PX was detected via immunoblot using HRP-conjugated anti-GST antibodies. For Fig EV4C, LUVs were generated as described previously (Ohashi *et al*, 2020). 100 nM of complex I (WT, 6SA or 6SE) was used for the assays. The activity was measured in the presence of 50 μM ATP using ADP-Glo assay kit (Promega). For each experiment, ATP-ADP mixtures were included to generate an ATP-ADP conversion line. The luminescence values were recorded by a microplate reader (PHERAstar, BMG Labtech). The values were subtracted by the intercept of the ATP-ADP conversion line (background subtraction). Specific activity was calculated as: background-subtracted luminescence value/slope of the conversion line/protein concentration/reaction time (min).

### Lipid flotation assays

Lipid flotation assays using LUVs were described previously (Ohashi *et al*, 2020).

### Western blot and immunofluorescence quantification

Image J (NIH) was used to quantify Western blots. For immunofluorescence analysis, WIPI2 puncta number was quantified using Imaris 8 × 64 software (Bitplane), with cell numbers counted manually.

### Statistical analysis

Experiments were repeated independently ≥ 3 times when quantifications are shown and ≥ 2 times if not, unless otherwise stated. Where WIPI2 puncta were counted, eight images (around 100 cells total) were analysed per condition per repeat. Mass spectrometry experiments were performed once only. GraphPad Prism 8.4.1 was used to plot graphs and calculate statistics. One-tailed ANOVAs were performed in all cases other than Figs 4F and, EV4C and D in which Student's *t*-tests were employed.

### Use of the ULK1 motif

In Fig 1B, the ULK1 motif (Egan *et al*, 2015) was rendered as follows, beginning at position −3 and ending at position +2 relative to the phosphoacceptor at position 0 (serine or threonine), with conforming residues in square brackets and non-conforming residues in braces: [M/L/Q/F]-{P}-{P}-[S/T]-[Y/I/M/S/F/V/W]{P}-[Y/S/H/I/W/M]{P}.

### Comment on nomenclature

Human protein/gene names are represented in uppercase, with the first letter along capitalised for non-human homologs (murine unless stated). Exceptions are human proteins Beclin1 and p62, spelled thus to correspond with the prevalent naming convention. "ULK" is used to refer to ULK homologs generally, typically ULK1/Ulk1 and ULK2/Ulk2. When referring to *S. cerevisiae* proteins, their gene names are given with the first letter capitalised and followed by the letter p.

## Data availability

The SILAC and TMT proteomics data sets are deposited at PRIDE hosted at the EBI (https://www.ebi.ac.uk/pride/archive/projects/PXD022228/). Data are available via ProteomeXchange with identifier PXD022228.

**Expanded View** for this article is available online.

### Acknowledgements

We thank Sila Ultanir (Francis Crick Institute) for helpful comments and advice, and David Carling, Naveenan Navaratnam (Imperial College), Shuyang Chen and Jon Wilson (The Francis Crick Institute) for their insight regarding analysis of PRKAG2 phosphorylation. Dhira Joshi and Nicola O'Reilly (Peptide Chemistry STP, Francis Crick Institute) and the Genomics Equipment Park (Francis Crick Institute) for contributing their invaluable expertise. Ok-Ryul Song and Michael Howell (High Throughput Screening STP, Francis Crick Institute) with image acquisition/scripting for the HTS analysis. Olga Perisic prepared mammalian cell cultures for VPS34 complex I purification and Saulé Spokaite helped Y.O. purify proteins. T.J.M., S.B., H.B.J.J., S.D.T., H.F., W.Z., M.W. D.F., A.P.S. and S.A.T

were supported by The Francis Crick Institute which receives its core funding from Cancer Research UK (FC001187, FC001999), the UK Medical Research Council (FC001187, FC001999). This research was funded in whole, or in part, by the Wellcome Trust (FC001187, FC001999). For the purpose of Open Access, the author has applied a CC BY public copyright licence to any Author Accepted Manuscript version arising from this submission. Y.O, S.T. and R.L.W were supported by the UK Medical Research Council [MC_U105184308 to R.L.W.] and Cancer Research UK (grant C14801/A21211 to R.L.W.).

## Author contributions

TJM performed cell and biochemical studies. HBJJ produced samples for the SILAC screen, established primary ULK DKO MEF cells and generated samples for high throughput screening imaging. SDT performed high throughput screening imaging and analysis. WZ produced 800 nm LUVs, SB performed the bioinformatics analyses, HF, DF and APS the mass spectrometry studies. YO, ST and RLW designed and performed VPS34 *in vitro* reconstitution assays and MW provided advice. TJM and SAT wrote the manuscript. All authors discussed the results and commented on the manuscript. TJM and SAT devised the experiments and supervised the work.

## Conflict of interest

The authors declare that they have no conflict of interest.

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
