## [Review Process File · The EMBO Journal]

Phosphoproteomic identification of ULK substrates reveals VPS15-dependent ULK/VPS34 interplay in the regulation of autophagy

Thomas Mercer, Yohei Ohashi, Stefan Boeing, Harold Jefferies, Stefano De Tito, Helen Flynn, Shirley Tremel, Wenxin Zhang, Martina Wirth, David Frith, Ambrosius P. Snijders, Roger Williams, and Sharon Tooze

DOI: [10.15252/embj.2020105985](https://doi.org/10.15252/embj.2020105985)

Corresponding author(s): Sharon Tooze (Sharon.tooze@crick.ac.uk)

Review Timeline:

Submission Date:	21st Jun 20
Editorial Decision:	31st Jul 20
Revision Received:	28th Oct 20
Editorial Decision:	23rd Nov 20
Revision Received:	12th Mar 21
Editorial Decision:	23rd Mar 21
Revision Received:	29th Mar 21
Accepted:	21st Apr 21

Editor: Elisabetta Argenzio

Transaction Report:

Thank you for the outline of the experiments that you have envisioned to address the referees' points. I have now discussed your action plan with the other members of the editorial team.

We appreciate you proposing to streamline the main text, test VPS15 phospho-S861 levels in WT and Ulk1/2 DKO MEFs, and further investigate the physiological role of VPS15 phosphorylation in autophagy initiation at the mechanistic level. However, we also notice that the outcome of these experiments cannot be predicted.

Even if there is a high degree of uncertainty, we offer you the opportunity to revise the manuscript as indicated in the reviews. I would point out that addressing all the referees' issues in a conclusive manner (particularly those regarding the physiological and mechanistic role of VPS15 phosphorylation in autophagy), as well as a unanimous strong support from the referees, will be essential for publication in The EMBO Journal.

Please note that it is The EMBO Journal policy to allow only a single major round of revision and it is therefore important that you resolve all of the main concerns at this stage.

We generally grant three months as standard revision time. As we are aware that many laboratories cannot function at full capacity owing to the COVID-19 pandemic, we may relax this deadline. Also, we have decided to apply our 'scooping protection policy' to the time span required for you to fully revise your manuscript and address the experimental issues highlighted herein. Nevertheless, please inform us as soon as a paper with related content published elsewhere.

When preparing your letter of response to the referees' comments, please bear in mind that this will form part of the Review Process File and will therefore be available online to the community. For more details on our Transparent Editorial Process, please visit our website:
http://emboj.embopress.org/about#Transparent_Process

Before submitting your revised manuscript, deposit any primary datasets (and computer code, where appropriate) produced in this study in an appropriate public database (see <http://msb.embopress.org/authorguide#dataavailability>). Please remember to provide a reviewer password, in case the datasets are not yet public. The accession numbers and database names should be listed in a formal "Data Availability" section (placed after Materials & Method). Provide a "Data availability" section even if there are no primary datasets produced in the study.

Finally, I would like to remind you that EMBO reports is still interested in your study. Therefore, you might consider it as an alternative venue for the rapid publication of your work. Martina Rembold (martina.rembold@embo.org) at EMBO reports has already annotated the referees' reports and would be happy to discuss with you the minimal requirement for publication in case you choose to transfer your manuscript there.

I thank you for the opportunity to consider this manuscript and look forward to hearing from you.

Referee #1:

The manuscript by Mercer et al. sets out to understand the role of ULK proteins in macroautophagy by identifying substrates for its kinase domain. Employing a series of complex phosphoproteomics, peptide arrays and more targeted approaches, they identify VPS15 as an ULK substrate. The authors dissect the effect of VPS15 and its phosphorylation on autophagy further by using a plethora of in vitro and cell-based approaches. Their data suggest that the phosphorylation of VPS15 by ULK promotes autophagosome formation. The manuscript provides a wealth of

additional information including a list of ULK substrates, a characterization of a novel mutant in the pseudokinase domain of VPS15 and report the accumulation and distribution of ULK substrates upon deletion of VPS15. Therefore, the manuscript is potentially interesting for a wider audience. As evident from the individual points below, this reviewer feels that the manuscript is very long, detailed and sometimes lacks some focus to make it intelligible for a broader readership.

1. Figure 2, page 8: the authors conduct kinase assays on peptide arrays, where additional S/T residues apart from the acceptor site were mutated to alanine. Do these mutations induce artifacts? Did the authors test this comparing the mutated peptide to the wt version?
2. Page 9: the authors conclude from their peptide array experiments "Based on this, we considered that peptide phosphorylation efficiency did not correlate perfectly with that in cells and probably did not reflect physiological relevance. Therefore, the results were considered qualitative and candidates were selected for further study based on feasible roles in autophagy." After fairly detailed description it is unclear to the general reader what to take home from these experiments.
3. The authors used MEFs for the proteomics experiments but HEK293A cells to delete VPS15 by CRISPR/Cas9. In HEK293A cells VPS15 is apparently essential for survival (page 11), whereas VPS15 KO MEFs exist (Ref 74). Why did the authors choose to use the HEK293 cells given that this choice complicates their studies and interpretations?
4. Figure 4E: the differences between the number of WIPI2 spots/cell in the cells rescued with WT VPS15 and the 6SA and 6SE mutants are small. In addition, it is doubtful that there is a significant difference between the 6SA and 6SE mutants. How sure can the authors be that the reduction of WIPI2 spots and by implication VPS34 activity isn't due to a reduced stability of the complex, in particular given that there is a slight destabilization of the PI4K complex for the 6SA mutant (Fig. 5A, Suppl. Fig S4).
5. This reviewer doesn't quite understand the experiment shown in Fig. 5B. Why is there an apparent difference in the activity of the wt and S6A PI3K C1 complexes, even though they were not phosphorylated by ULK1, and why was the S6E mutant not included? Doesn't this experiment point to an unspecific, phosphorylation independent effect of the S6A mutant? From other studies it is clear that phosphorylation of PI3K C1 by ULK1 is not required for its lipid kinase activity (PMIDs: 32437499, 32602837).
6. The sentence "Exemplifying this, for some of the selected substrates, most if not all of the identified phosphopeptides were depleted in DKO (e.g. see Sorbs2 in Supplemental Table 1), indicative of protein level variation rather than loss of ULK-dependent phosphorylation." on page 7 is somewhat unclear to this reviewer. Can the authors please rephrase?
7. Some of the description of the phosphoproteomics analysis is very detailed and sometimes it is difficult to grasp what the actual message is. The authors may want to consider moving some of these details to the methods. Similar, the description of the generation of the VPS15 KO/depleted cell lines on page 11 is rather long and complex.
8. The characterization of the deltaV50 VPS15 mutant is interesting. However, it is not directly related to the main story/title of the paper, which is the characterization ULK1 substrates. It is of course up to the authors, but it may distract the reader from the main message. Related to this, shouldn't the title read "INSIGHTS INTO THE Regulation of Autophagy via Identification and Characterisation of ULK Kinase Substrates", as the identification and characterization itself doesn't

regulate autophagy.

9. Figure 4B: because the quantification is based on only 2 experiments, it would be better to show the two data points rather than columns with error bars. Also, it should be clearly indicated what the different greys indicate.

10. Page 10: the authors write "Interestingly, although well conserved across multiple lineages, these intrinsically disordered regions are poorly conserved in *S. cerevisiae*.". It would be helpful to show an alignment of the relevant region of VPS15 to show its conservation in other species.

11. Page 13: the sentence "Phosphorylation of ULK substrate S1289 was not required, however it may control Y1290 phosphorylation (Supplemental Figure 4E, Supplementary Table 4)." is confusing. The authors should formulate more precisely what the ULK1 substrate S1289 is not required for.

12. The authors decided to focus on VPS15 as a novel substrate of ULK. In Suppl. Table 2 it is listed as PIK3R4, which could be confusing to some readers. The authors may want to harmonize the nomenclature used. Likewise, on page 13 the name p150 is used.

Referee #2:

The ULK1/2 kinase complex initiates the autophagy signaling cascade in response to various stress conditions. Many ULK substrates have been reported, but the significance of each phosphorylation to the substrate's function and autophagy regulation is only partially understood. In the present study, the authors obtain a non-biased phosphoproteome of ULK1/2DKO cells and identify many ULK-dependent phosphorylation sites. Amongst the identified proteins, they focus on VPS15 as a novel ULK substrate and propose that its phosphorylation is important for autophagy regulation. Overall, although the systematic identification of ULK substrates is much appreciated, the importance of VPS15 phosphorylation to autophagy regulation is not evident and should be clarified.

Major concerns

1. That ULK phosphorylates VPS15 is unsurprising since it's already known to phosphorylate other subunits of VPS34 complex I (i.e., VPS34, Beclin 1, ATG14, and NRBF2). Nevertheless, the novelty of this study could have been enhanced if VPS15 phosphorylation was found to be essential to autophagy regulation. This doesn't seem to be the case. The phospho-deficient mutant VPS15-6SA shows only a 25% reduction in autophagic flux (Fig. 4) and virtually no phenotype was observed for the S861A mutant (Fig. 6F). More confusingly, the phospho-mimic mutant VPS15-6SE can restore autophagic flux in VPS15KO cells (Fig. 4), whereas the S861E mutant rather inhibits autophagy (Fig. 6). How VPS15 phosphorylation regulates the kinase activity of VPS34 is also not investigated. In summary, VPS15 phosphorylation does not appear to be physiologically important (even though it is a newly identified substrate).

2. The authors generated an antibody for phospho-VPS15 but did not check whether its phosphorylation actually depends on endogenous ULK (which can be determined by comparing

wild-type and ULK1/2DKO cells).

3. The authors seem to have confused the role of VPS15 and the role of phosphorylated VPS15. For example, the data shown in Fig. 7 are not relevant to VPS15 phosphorylation and are mostly predictable from the findings of previous studies that characterized the other subunits of VPS34 complex I. The phenotype of VPS Δ V50 may be interesting, but how it's related to ULK-dependent phosphorylation is not clear. The findings on VPS Δ V50 may be better presented as an independent study.

Minor concerns

1. The authors refer to the ULK complex as a "tetrameric" complex in Introduction. This is inaccurate as FIP200 dimerizes and a recent paper suggests that this complex is pentameric (PMID: 32516362).

2. "Figure 4G" should be "Figure 4F".

3. It is not clear to this reviewer why the authors conclude that VPS Δ V50 has a dominant-negative effect when it doesn't inhibit autophagy in wild-type cells.

Referee #3:

Mercer et al., performed a phospho-proteomic screen associated to a peptide array screen to identify new ULK1 substrates. From the comparison of the screen authors moved their attention to VPS15, which is a component of the bigger complex the class III PI3Kinase. Authors identified the Ser phosphorylated by ULK1, investigated the functional role of the phosphorylation in term of PI3K complex stability and autophagy induction. Moreover, from their CRISPR Cas9 gene deletion they highlight the essential function of VPS15 in HEK293A cells and how VPS15 KO determines the accumulation of ULK phospho-substrates and formation of aberrant structures. Moreover, the CRISPR clones show the importance of Valine 50 removal for the biological function of VPS15. The manuscript adds a new brick in the wall of PI3K complex and unveil the role of ULK1-VPS15 in autophagy. It is also interesting how a single aa deletion (Val 50) could be so detrimental for VPS15 function.

Authors performed a huge amount of experiments and data amount is relevant. However, sometimes it's difficult to follow the text and authors' thoughts. This makes the manuscript lengthily and sometimes difficult to properly understand. I would suggest a general editing of the text reducing the amount of information and details (some can be simply omitted). Twenty-five pages of body text and more than a hundred references maybe is a bit too much for a research article.

- The first part is dedicated to the phospho-proteomic and peptide array screens. This part should be simplified maybe introducing some schemes to lead the readers. Authors described at least three different approaches: two unbiased (proteomics) and one more restricted (peptide arrays). Moreover, authors introduced several restrictions in data analysis to reduce the amount of hits (e.i consensus aa sequences). This is understandable, but it also adds some limits. I would suggest reducing the body text in order to make it easier to follow and better specify why several biased have been introduced.

- Many controls and/or less relevant data can be moved to supplementary and text can be shorted
- The CRISPR part is also sometime confusing. The authors wrote that "deletion of valine 50 for the

majority of alleles correlates with a KO phenotype better than VPS15 depletion." This sentence is a bit misleading. Deletion of Val 50 should determine a dominant negative effect. Moreover, the fact that HEK cells have multiple alleles with VPS15 could be a resource considering the essentiality of the gene but also a problem because the system seems to be quite heterogeneous. I would suggest reproducing data in a different cellular system in order to clarify this point.

- Please try to avoid "data not shown" especially for experiments related to VPS15 phosphorylation or Val 50 deletion.

- Autophagy flux analyses should not be limited to a single time point.

In this format the manuscript is quite difficult to follow. Considering that the main message regards VPS15 and its phosphor-regulation mediated by ULK1, I would suggest to focus on that and leave many additional information that are not really necessary for this topic.

Response to referees
Mercer et al.,
EMBO J 12020-105985

We thank the referees for their comments, which were fair in their appraisals and are pleased that they recognised the wide interest of our study. Importantly, we believe that the comments they made, all of which we address, have improved the manuscript substantially. To streamline the text, we have made three major changes. We have pared down the data on the VPS15 KOs by removing unnecessary controls as requested by Reviewers #2 and #3 and removed the majority of data relating to the $\Delta V50$ phenotype. The latter ran alongside study of the phosphorylation-dependent phenotype so distracted from the main message as noted by Reviewers #1 and #2. We also simplified our description and discussion of our screening steps and distributed parts to the methods section. Alongside the major changes, we have removed all instances where 'data not shown' were discussed, and clarified specific sentences highlighted by the referees. These changes have shortened the manuscript and brought into focus the importance and major conclusions of our work, which are now intelligible for a broader readership.

In removing $\Delta V50$ data, we improved the focus the manuscript and thus present a comprehensive study of ULK-dependent signalling. We include additional data to this end: we have performed new peptide array-based *in vitro* kinase assays in which the Ulk1 and Ulk2 complexes were used to phosphorylate the high confidence list of substrates. These replace previous data in which a catalytically active fragment of Ulk1 was used. Using the more physiologically relevant full complexes improved confidence in the screen results (evidenced by phosphorylation of an expanded range of positive controls) and increased the number of validated substrates. These experiments offer the first unequivocal evidence that Ulk1 and Ulk2 have almost identical *in vitro* specificities. They also allowed us to pare down the lengthy discussion.

Despite numerous attempts we were unable to identify the physiological stimulus for VPS15 S861 phosphorylation (please see attached figure for referees, panel B). However, our most recent findings uncover the physiological stimuli governing phosphorylation of the ULK substrates in two key autophagic signalling complexes: PRKAG2 S124 and VPS34 S249. PRKAG2 S124 is phosphorylated by both ULK and AMPK, and is highly sensitive to serum status. For VPS34 S249, we showed that VPS15 depletion, amino acid starvation and the mitophagy-inducing iron chelator deferiprone strongly promotes phosphorylation using the commercial phosphoantibody (previously reported to work only upon ULK and VPS34 co-overexpression). We stress the interest of these new findings which are in keeping with the main aim of this study, to understand the mechanisms of autophagy by studying the biology of the ULK kinases, primarily via the identification and characterisation of novel substrates.

As requested, we tested how autophagic flux is modulated in phosphomutant-rescued cells over time to strengthen the role of VPS15 phosphorylation in autophagy initiation (Figure 6C). We also examined how ULK-dependent VPS15 phosphorylation controls VPS34 kinase activity and PI3P production by testing the relative *in vitro* kinase activities of complexes containing wild type, 6SA and 6SE VPS15 (Supplemental Figure 4A). Together, these data further illuminate how VPS15

phosphorylation is regulated and provide additional mechanistic insight into the downstream consequences and physiological role of VPS15 phosphorylation.

We believe the data presented in the manuscript address the physiological relevance of the ULK-VPS15 signalling axis but in the original submission may have been overshadowed less relevant information which we have now removed. The SILAC screen provided evidence for ULK-dependent VPS15 phosphorylation upon starvation in endogenous settings, and *in vitro* reconstitution experiments revealed that mutating the ULK substrate residues in VPS15 significantly reduces VPS34 complex activity, possibly via a reduction in membrane binding. Furthermore, stable phosphorylation of S861 alone, which is depleted in endogenous VPS15 when ULK kinases are removed, significantly reduced autophagy in rescued cells without destabilising the VPS34 complex (see attached data, panel A).

We also want to stress the physiological relevance of the data presented in Figures 7 and 8. We have added new data showing that the accumulation of aberrant structures upon VPS15 ablation is a VPS34 complex I-dependent phenotype, overturning current understanding of the VPS15 KO phenotype. Furthermore, the novel and striking ULK-dependent phenotypes that occur upon prolonged VPS34 inhibition provide intriguing insights into the complexity of cargo-dependent ULK-complex recruitment as recently expounded by the Martens, Youle and Randow laboratories.

Figures Added (Using new Figure numbering)

Figure 1F – Tightens focus on ULK substrates in VPS34 and AMPK complexes to aid flow into next section

Figure 2D – Upstream stimuli governing PRKAG2 S124 phosphorylation revealed

Figure 6C – Autophagic flux analysis over time course, as requested by Reviewer 3. Also shows progressive accumulation of VPS34 S249 phosphorylation indicative of nutrient-dependent phosphorylation

Figure 8C – Improvement on previous data, FIP200 had same distribution as ULK1 but antibody better suited to immunofluorescence. Included quantification of FIP200 body number showing significant increase in ATG14 KO condition.

Figure EV2 – performed new peptide array-based *in vitro* kinase assay using full Ulk1 and Ulk2 complexes. Improved physiological relevance and reliability evidenced by phosphorylation of a wider range of positive control and experimental candidates

Figure EV4A – Assessed *in vitro* lipid kinase activity of VPS34 complex I incorporating VPS15 6SA and 6SE on 800nm LUVs

Figure EV5B, C, D – Upstream stimuli governing VPS34 S249 phosphorylation revealed

Figures Removed (Using old Figure numbering)

Figure 4D – As delayed endolysosomal trafficking is an established phenotype of PI3P depletion, control for VPS15 knockout phenotype considered superfluous

Figure 5D, E, F – VPS15 Δ V50 phenotype no longer included

Supplemental Figure 3E – Discussion of phenotypic differences between CRISPR clones and how this relates to their genotypes no longer relevant

Supplemental Figure 4C, D – VPS15 Δ V50 phenotype no longer included

Supplemental Figure 4E-Characterisation of phosphorylation-regulated LIR in VPS15 not directly relevant

Supplemental 5-incorporated into Figure 4 (now Figure 4E and 4F)

Supplemental Figure 6C – Panel superfluous, similar data in separate VPS15 KO clone shown in panel (Now Figure 6E)

Supplemental Figure 6E – Replaced with quantified IF images in which ULK complex was revealed by staining for FIP200 rather than ULK1 (antibody better suited for IF), now Figure EV8C.

Data included for reviewers though excluded from paper

- Panel A: HEK293A coexpressing VPS34 along with empty vector (EV) or with wild type (WT), S861A or S861E VPS15 were starved for 30 minutes before lysis and VPS15 immunoprecipitation. Western blot analysis was used to assess coimmunoprecipitation of VPS34 complex I and II core components. Quantification of 4 independent repeats revealed no significant differences in complex member binding.
- Panel B: Multiple autophagy-inducing treatments were screened for capacity to activate ULK. Most promising agonists were used to treat HEK293A overexpressing VPS34 complex I to near-endogenous levels. Complexes were immunoprecipitated via ATG14-ZZ and VPS15 pS861 probed by Western blot. As a control, immunoprecipitated VPS34 complex I was phosphorylated *in vitro* by Ulk1 to enrich serine 861 phosphorylation before treatment with lambda phosphatase to remove phosphorylation.

Specific Answers to Referee's Comments

Referee #1

The manuscript by Mercer et al. sets out to understand the role of ULK proteins in macroautophagy by identifying substrates for its kinase domain. Employing a series of complex phosphoproteomics, peptide arrays and more targeted approaches, they identify VPS15 as an ULK substrate. The authors dissect the effect of VPS15 and its phosphorylation on autophagy further by using a plethora of *in vitro* and cell-based approaches. Their data suggest that the phosphorylation of VPS15 by ULK promotes autophagosome formation. The manuscript provides a wealth of additional information including a list of ULK substrates, a characterization of a novel mutant in the pseudokinase domain of VPS15 and report the accumulation and distribution of ULK substrates upon deletion of VPS15. Therefore, the manuscript is potentially interesting for a wider audience.

As evident from the individual points below, this reviewer feels that the manuscript is very long, detailed and sometimes lacks some focus to make it intelligible for a broader readership.

We thank the referee for their recognition of the contribution our manuscript makes, and in particular to a wider audience.

1. Figure 2, page 8: the authors conduct kinase assays on peptide arrays, where additional S/T residues apart from the acceptor site were mutated to alanine. Do these mutations induce artifacts? Did the authors test this comparing the mutated peptide to the wt version?

Apologies for any lack of clarity. In the screen, WT, single phosphoacceptor (predicted site), total phosphoacceptor (additional S/T residues) and, where relevant, murine sequences were compared. We have restructured this section and performed new experiments as described above.

2. Page 9: the authors conclude from their peptide array experiments "Based on this, we considered

that peptide phosphorylation efficiency did not correlate perfectly with that in cells and probably did not reflect physiological relevance. Therefore, the results were considered qualitative and candidates were selected for further study based on feasible roles in autophagy." After fairly detailed description it is unclear to the general reader what to take home from these experiments.

We were attempting to explain that phosphorylation efficiency in our assay does not need to conform with the potential importance of the phosphorylation for ULK-dependent autophagy. We have removed this sentence for clarity.

3. The authors used MEFs for the proteomics experiments but HEK293A cells to delete VPS15 by CRISPR/Cas9. In HEK293A cells VPS15 is apparently essential for survival (page 11), whereas VPS15 KO MEFs exist (Ref 74). Why did the authors choose to use the HEK293 cells given that this choice complicates their studies and interpretations?

In our study, we identify ULK-dependent phosphorylation of VPS15 in two cell lines, strengthening confidence in its identification. We chose HEK293A cells for validation as it is a well characterised human cell model in which to study autophagy, for which we possess most of the required tools and expertise. We showed that HEK293A are polyploid with respect to VPS15, however we used cells expressing two alleles only (dV50 and WT), and in all rescue experiments exogenous VPS15 was expressed at high enough levels to dominate and overcome any impact of the mutant endogenous genes. We believe that HEK293A polyploidy uniquely facilitated generation of VPS15 hypomorphs as we were able to mutate the majority of alleles whilst retaining a small fraction unmodified.

Importantly, we note that the VPS15 KO MEFs are missing only the first coding exon. They express a truncated form of VPS15, lacking the pseudokinase domain only, to normal levels and are therefore not a comparable model and are less suitable for rescue experiments.

4. Figure 4E: the differences between the number of WIPI2 spots/cell in the cells rescued with WT VPS15 and the 6SA and 6SE mutants are small. In addition, it is doubtful that there is a significant difference between the 6SA and 6SE mutants. How sure can the authors be that the reduction of WIPI2 spots and by implication VPS34 activity isn't due to a reduced stability of the complex, in particular given that there is a slight destabilization of the PI4K complex for the 6SA mutant (Fig. 5A, Suppl. Fig S4).

For the 6SA phenotype, we note that the slightly reduced pulldown of VPS34 with VPS15 6SA may reflect a physiological function of ULK-dependent phosphorylation (i.e. ULK phosphorylation promotes stability of the VPS34 complex). However, the VPS34 complexes used for *in vitro* lipid kinase assays (Figure 4E and 4F) which were more thoroughly purified from cells demonstrated virtually no difference in stability yet a contrastingly large reduction in kinase activity. Of note, cells were starved before lysis in Supplemental Figure 4B, but not for Figure 4E-G.

We also highlight that no difference in VPS34 complex member pulldown was observed between WT, S861A and S861E-containing complexes despite the similar reduction in autophagic flux/WIPI2 puncta formation in 6SA and S861E-expressing cells (please see attached figure for referees, panel A).

Together, we suggest that our data might reflect multiple phenotypic consequences depending on which sites in VPS15 are phosphorylated by ULK. Our S861E rescue data show that phosphorylation of this site reduces starvation-induced autophagy and we think it is likely that this site alone could be responsible for the 6SE phenotype. Alternatively, it is possible that the serine to glutamate mutation is insufficient in mimicking phosphorylation and thus is effectively a non-phosphorylatable form of VPS15, which is very hard to test and may not add any useful information. Importantly, our data consistently demonstrate that autophagy/VPS34 activity is modulated when ULK target residues are mutated.

5. This reviewer doesn't quite understand the experiment shown in Fig. 5B. Why is there an apparent difference in the activity of the wt and S6A PI3K C1 complexes, even though they were not phosphorylated by ULK1, and why was the S6E mutant not included? Doesn't this experiment point to an unspecific, phosphorylation independent effect of the S6A mutant? From other studies it is clear that phosphorylation of PI3K C1 by ULK1 is not required for its lipid kinase activity (PMIDs: 32437499, 32602837).

We propose that a proportion of VPS15 WT in the VPS34 complexes purified for these experiments was basally phosphorylated at one or a combination of the 6 newly identified sites. The referenced papers indicate that VPS34 lipid kinase activity can be modulated by introducing cofactors/altering membrane characteristics, we show that baseline lipid kinase activity is reduced by blocking ULK phosphorylation (Figure 4F). Supporting these points, serine to alanine mutation of the ULK1 substrate ATG14 serine 29 was shown previously to reduce *in vitro* lipid kinase activity in basal conditions (Park et al., Autophagy 2016, PMID 27046250).

6SE VPS15 was not included initially in reconstitution experiments as we saw the largest phenotype in cells rescued with VPS15 6SA.

6. The sentence "Exemplifying this, for some of the selected substrates, most if not all of the identified phosphopeptides were depleted in DKO (e.g. see Sorbs2 in Supplemental Table 1), indicative of protein level variation rather than loss of ULK-dependent phosphorylation." on page 7 is somewhat unclear to this reviewer. Can the authors please rephrase?

Clarified sentence, we hope this has addressed your concern.

7. Some of the description of the phosphoproteomics analysis is very detailed and sometimes it is difficult to grasp what the actual message is. The authors may want to consider moving some of these details to the methods. Similar, the description of the generation of the VPS15 KO/depleted cell lines on page 11 is rather long and complex.

We agree and have streamlined the text as described above.

8. The characterization of the deltaV50 VPS15 mutant is interesting. However, it is not directly related to the main story/title of the paper, which is the characterization ULK1 substrates. It is of course up to the authors, but it may distract the reader from the main message. Related to this, shouldn't the title read "INSIGHTS INTO THE Regulation of Autophagy via Identification and Characterisation of ULK Kinase Substrates", as the identification and characterization itself doesn't regulate autophagy.

We agree on both points and have both removed the majority of $\Delta V50$ data and changed the title.

9. Figure 4B: because the quantification is based on only 2 experiments, it would be better to show the two data points rather than columns with error bars. Also, it should be clearly indicated what the different greys indicate.

We have included further experimental repeats and fixed the colour scheme on the graph.

10. Page 10: the authors write "Interestingly, although well conserved across multiple lineages, these intrinsically disordered regions are poorly conserved in *S. cerevisiae*". It would be helpful to show an alignment of the relevant region of VPS15 to show its conservation in other species.

Statement rephrased and reference to multiple sequence alignment provided (see figure 3C in reference 39; Rostislavleva et al., Science 2015, PMID 26450213).

11. Page 13 (now page 11): the sentence "Phosphorylation of ULK substrate S1289 was not

required, however it may control Y1290 phosphorylation (Supplemental Figure 4E, Supplementary Table 4)." is confusing. The authors should formulate more precise what the ULK1 substrate S1289 is not required for.

The data addressing the role of S1289 has been removed to improve the focus of the manuscript.

12. The authors decided to focus on VPS15 as a novel substrate of ULK. In Suppl. Table 2 it is listed as PIK3R4, which could be confusing to some readers. The authors may want to harmonize the nomenclature used. Likewise, on page 13 the name p150 is used.

Consistent nomenclature now used throughout. Note that Supplementary Table 1 is exception, but an explanation has been added to legend.

Referee #2

The ULK1/2 kinase complex initiates the autophagy signaling cascade in response to various stress conditions. Many ULK substrates have been reported, but the significance of each phosphorylation to the substrate's function and autophagy regulation is only partially understood. In the present study, the authors obtain a non-biased phosphoproteome of ULK1/2DKO cells and identify many ULK-dependent phosphorylation sites. Amongst the identified proteins, they focus on VPS15 as a novel ULK substrate and propose that its phosphorylation is important for autophagy regulation. Overall, although the systematic identification of ULK substrates is much appreciated, the importance of VPS15 phosphorylation to autophagy regulation is not evident and should be clarified.

We thank the referee for their appreciation of the phosphoproteomics and we have improved the manuscript to address not only the importance of VPS15 phosphorylation but also the role of ULK in the regulation of VPS34.

Major concerns

1. That ULK phosphorylates VPS15 is unsurprising since it's already known to phosphorylate other subunits of VPS34 complex I (i.e., VPS34, Beclin 1, ATG14, and NRBF2). Nevertheless, the novelty of this study could have been enhanced if VPS15 phosphorylation was found to be essential to autophagy regulation. This doesn't seem to be the case. The phospho-deficient mutant VPS15-6SA shows only a 25% reduction in autophagic flux (Fig. 4) and virtually no phenotype was observed for the S861A mutant (Fig. 6F). More confusingly, the phospho-mimic mutant VPS15-6SE can restore autophagic flux in VPS15KO cells (Fig. 4), whereas the S861E mutant rather inhibits autophagy (Fig. 6). How VPS15 phosphorylation regulates the kinase activity of VPS34 is also not investigated. In summary, VPS15 phosphorylation does not appear to be physiologically important (even though it is a newly identified substrate).

We stress that there are several novel aspects of our study. Alongside the wide range of new substrates, we identify VPS15 and UVRAG, showing the full complexity of ULK-VPS34 signalling by revealing all VPS34 complex components as ULK substrates. We identify multiple stimuli for VPS34 S249 and PRKAG2 S124 phosphorylation. On top of this we are the first group to characterise phosphorylation sites in VPS15. Whilst phosphorylation of the 6 novel substrates is not essential for autophagy, we note that the vast majority of the ULK substrate residues identified to date are not essential for autophagy regulation. While it may be unsurprising that ULK phosphorylates VPS15 our data demonstrating that this is in fact true is required for progress in understanding the biology of the VPS34 complex.

We believe that our data may reveal multiple phenotypic consequences downstream of the ULK-VPS15 signalling axis depending on which sites in VPS15 are phosphorylated by ULK. S861 phosphorylation is largely inhibitory to VPS34 activation, however this may be overcome by phosphorylation of the other 5 sites alone or in combination. We are so far unable to identify the stimulus for VPS15 (see Panel B in provided data); however, we stress that its physiological relevance may become clear in the future. Exemplifying this, when VPS34 S249 was identified as an ULK substrate, its function and regulation could not be determined (Egan et al., Mol Cell 2015, PMID 26118643). However, the phosphomimic S249E was recently shown to generate a LIR and thus drive LC3/GABARAP protein association (Birgisdottir et al., Autophagy 2019, PMID 30767700), which we recapitulated using peptides bearing phosphoserine at serine 249 (see panel S4E from previous submission, now removed). In identifying multiple stimuli, our new data now provide the previously elusive physiological context.

2. The authors generated an antibody for phospho-VPS15 but did not check whether its

phosphorylation actually depends on endogenous ULK (which can be determined by comparing wild-type and ULK1/2DKO cells).

We highlight that S861 was identified in the SILAC screen as it was among the most highly depleted phosphopeptides in ULK1/2 DKO MEFs. This experiment used endogenously expressed proteins so were strongly indicative of ULK-dependent phosphorylation.

Based on your input we have attempted to recapitulate these findings using our rabbit phosphoantibody. Unfortunately, it has a very low titer and affinity and requires enrichment of the epitope before a reliable signal can be detected. We therefore attempted to pull VPS15 out of MEF lines, but we couldn't immunopurify the VPS34 complex from MEFs reliably using a non-rabbit antibody. Alternatively, we attempted to show endogenous ULK-dependent phosphorylation in HEK293A overexpressing VPS34 complex members to near-endogenous levels. The final results of these experiments are attached (see Panel B in provided data). In summary, we were not able to identify the stimulus, but believe this could be achieved in future studies with improved materials for detection of S861 phosphorylation.

3. The authors seem to have confused the role of VPS15 and the role of phosphorylated VPS15. For example, the data shown in Fig. 7 are not relevant to VPS15 phosphorylation and are mostly predictable from the findings of previous studies that characterized the other subunits of VPS34 complex I. The phenotype of VPS Δ V50 may be interesting, but how it's related to ULK-dependent phosphorylation is not clear. The findings on VPS Δ V50 may be better presented as an independent study.

We wish to stress that the aim of our report is to study the biology of the ULK kinases. The aberrant structures we detect are likely similar to those reported in multiple ATG KO models, which have been ascribed as aggregates, aberrant early autophagic structures and stalled autolysosomes by various groups. We connect the recent findings from the Martens, Youle and Randow laboratories to both support and expand on the model that the structures represent aggregate-associated early autophagic structures (similar to those proposed in Kishi-Itakura et al., J Cell Science, PMID 25052093) and to identify VPS15 KO cells as an unexpected model in which to study cargo-dependent ULK-complex recruitment.

In data added in this revision, we show that structure formation is complex I-dependent, challenging the previous understanding that the primary autophagic defect in VPS15 KO cells was due to prevention of autophagosome maturation. Importantly, the accumulation of ULK substrates and ULK activity-dependent redistribution of early autophagy proteins were unexpected, previously unreported and, in the latter case, currently unexplained. We believe that our data add novel insight into the mechanisms by which cargo act as platforms for autophagic signalling complexes.

We agree with the concerns regarding the Δ V50 data and have removed them accordingly.

Minor concerns

1. The authors refer to the ULK complex as a "tetrameric" complex in Introduction. This is inaccurate as FIP200 dimerizes and a recent paper suggests that this complex is pentameric (PMID: 32516362).

Correction incorporated into text

2. "Figure 4G" should be "Figure 4F".

Figure labelling amended

3. It is not clear to this reviewer why the authors conclude that VPS Δ V50 has a dominant-negative effect when it doesn't inhibit autophagy in wild-type cells.

This text has been removed

Referee#3

Mercer et al., performed a phospho-proteomic screen associated to a peptide array screen to identify new ULK1 substrates. From the comparison of the screen authors moved their attention to VPS15, which is a component of the bigger complex the class III PI3Kinase. Authors identified the Ser phosphorylated by ULK1, investigated the functional role of the phosphorylation in term of PI3K complex stability and autophagy induction. Moreover, from their CRISPR Cas9 gene deletion they highlight the essential function of VPS15 in HEK293A cells and how VPS15 KO determines the accumulation of ULK phospho-substrates and formation of aberrant structures. Moreover, the CRISPR clones show the importance of Valine 50 removal for the biological function of VPS15. The manuscript adds a new brick in the wall of PI3K complex and unveil the role of ULK1-VPS15 in autophagy. It is also interesting how a single aa deletion (Val 50) could be so detrimental for VPS15 function.

Authors performed a huge amount of experiments and data amount is relevant. However, sometimes it's difficult to follow the text and authors' thoughts. This makes the manuscript lengthily and sometimes difficult to properly understand. I would suggest a general editing of the text reducing the amount of information and details (some can be simply omitted). Twenty-five pages of body text and more than a hundred references maybe is a bit too much for a research article.

We thank the referee for their constructive summary and suggestions. We agree the large amount of data and the length of the text hindered our ability to convey the key findings. We have streamlined and edited the text to improve clarity and make it more concise and focused. We have not reduced the references as we feel strongly about the need to include relevant published data to provide the readers with the possibility to understand the literature.

- The first part is dedicated to the phospho-proteomic and peptide array screens. This part should be simplified maybe introducing some schemes to lead the readers. Authors described at least three different approaches: two unbiased (proteomics) and one more restricted (peptide arrays). Moreover, authors introduced several restrictions in data analysis to reduce the amount of hits (e.i consensus aa sequences). This is understandable, but it also adds some limits. I would suggest reducing the body text in order to make it easier to follow and better specify why several biased have been introduced.

We note that we did not use consensus AA sequences to triage the shortlisted hits and apologise for any confusion caused. We have clarified our description of screening stages and the logic behind them.

- Many controls and/or less relevant data can be moved to supplementary and text can be shorted

We have removed less relevant controls and data allowing us to significantly shorten the text. The data removed is listed above.

- The CRISPR part is also sometime confusing. The authors wrote that "deletion of valine 50 for the majority of alleles correlates with a KO phenotype better than VPS15 depletion." This sentence is a bit misleading. Deletion of Val 50 should determine a dominant negative effect. Moreover, the fact that HEK cells have multiple alleles with VPS15 could be a resource considering the essentiality of the gene but also a problem because the system seems to be quite heterogeneous. I would suggest reproducing data in a different cellular system in order to clarify this point.

We agree and have accordingly removed the misleading sentence. We note that despite the variety of VPS15 alleles in HEK293A, all of our effective knockouts reproduce the same phenotype despite slight variations in their genotype. Moreover, this phenotype is highly similar to the reported VPS34 KO MEF phenotype, and virtually identical to the VPS15 KO MEF phenotype reported in Nemazany et al., EMBO Mol Med 2013, PMID 23630012. We believed that any allele heterogeneity is overcome in rescued cells upon expression of WT/phosphomutant VPS15. Regarding an

alternative cellular system, as discussed above, we are concerned that the VPS15 KO MEFs are unsuitable for our experiments as they express the truncated form of VPS15. Together, these insights suggest that our model is sufficient to study the VPS15 KO and phosphomutant phenotypes.

- Please try to avoid "data not shown" especially for experiments related to VPS15 phosphorylation or Val 50 deletion.

We have removed all of instances of 'data not shown'.

- Autophagy flux analyses should not be limited to a single time point.

We have performed a time course of flux (new Figure 6C) and incorporated the resulting data. This solidifies our conclusions.

In this format the manuscript is quite difficult to follow. Considering that the main message regards VPS15 and its phosphor-regulation mediated by ULK1, I would suggest to focus on that and leave many additional information that are not really necessary for this topic.

We hope we have refocussed and clarified the main aim of the paper during revision.

A

B

Data for referees

Thank you for submitting your revised manuscript. The study has been sent back to the original referees for evaluation and we have now received their reports, which are enclosed below for your information.

As you can see, referee #3 finds that his/her concerns have been sufficiently addressed. However, reviewer #1 is not satisfied by your answers to his/her point 4 and 5 concerning the stability of the PI3K 6A mutant complex and its reduced activity in vitro. In addition, referee #2 states that, while determining the precise physiological function of each phosphorylation site in VPS15 may be beyond the scope of this study, determining whether phosphorylation inhibits or stimulates autophagy is crucial. Given that these criticisms are important and concern fundamental aspects of your study, I would invite you to address them as requested by the referees.

I thank you again for giving us the chance to consider your manuscript for publication in The EMBO Journal and look forward to your revision.

Referee #1:

The authors have addressed many of my comments adequately and the manuscript is easier to follow now. However, I am still not convinced by the answers to points 4 and 5 regarding the stability of the PI3K 6A mutant complex and its reduced activity in the in vitro assay.

1. The authors argue that the reduced lipid kinase activity of the 6A mutant complex compared to the wild type complex is due to basal phosphorylation of the wt PI3K complex rather than to the reduced stability or purity of the 6A mutant, which however seems evident from the gel shown in Fig. 4E. This hypothesis could easily be tested by incubating the purified wt PI3K complex with a phosphatase. This treatment should reduce its activity to that of the 6A mutant. In addition, it is not comprehensible to this reviewer why the 6E mutant was not included in the assay shown in Fig. 4F.
2. On page 10 the authors write "The chronic PI3P depletion phenotype observed in CRISPR clones sgA-A8, sgA-B3, sgA-D6 and sgA-D11 confirmed that they are effective VPS15 knockouts (now referred to as VPS15 KOs)". However, I understand that some VPS15 expression still exists in these cells as complete knock out is likely to be lethal. The authors should therefore rephrase this sentence as it could easily be misunderstood.
3. As a more minor point, the authors write on page 6 that ULK1 would preferentially phosphorylate interacting substrates citing a review (REF 20). This sentence should be clarified as evidently all substrates must at least transiently interact. In addition, rather than a review, the actual evidence should be cited.
4. On page 11, last sentence the word "showed" or "revealed" should be deleted.

Referee #2:

Despite multiple improvements, there are still some issues that should be clarified.

The added data in Fig. 6C would support the authors' claim that phosphorylation of S861 is inhibitory for autophagic flux were it not for the contradictory finding that S861A inhibits flux (Fig. 6B). The authors discuss that "whilst the 6SA rescue/reconstitution data indicate that the ULK-dependent VPS15 phosphorylation positively regulates autophagy, phosphorylation of S861 alone is largely inhibitory" without showing data and also not taking into account the data of 6SA and 6SE VPS15 mutants both showing inhibitory effects (Fig. 4D). As long as both SA and SE mutants demonstrate similar effects, it's difficult to tell how these phosphorylations may regulate autophagy. Determining the precise physiological function of each phosphorylation site may be beyond the scope of this study but determining whether these phosphorylations inhibit or stimulate autophagy is crucial. If S861 is a primary inhibitory site, the authors should at least test the effects of S5E and S5A mutants (with intact S861) to dissect the role of S861 phosphorylation from those of other phosphorylations.

The authors added new data demonstrating that FIP200-positive structures accumulate in siATG14 cells in a PI3K C1-dependent manner. However, it has been shown that Vps34 kinase activity regulates the dynamics of the ULK1 complex (e.g., in Ref #1, 21), which may reduce the novelty of the authors' new data. The authors should clarify the difference between these previous studies and the current study.

Referee #3:

In the revised version of the manuscript, the authors addressed all the points raised by the referees. Moreover, they edited the text that now is more compact and easier to follow. This referee doesn't have other major concerns regarding the manuscript. Maybe just an advice after reading the rebuttal letter. Page 7 of rebuttal letter, the authors wrote " Whilst phosphorylation of the understanding the biology of the VPS34 complex". Considering this sentence the author may include in the title a mention to VPS34 complex.

Summary of responses and changes:

Figures Removed

1. Figure 4E – Replaced with updated data
2. Figure 4F – Replaced with updated data
3. Figure 4G – Replaced with updated data
4. EV3 Figure E – removed to streamline manuscript

Figures Added

1. Figure 4E – Included to address reviewer 1 point 1
2. Figure 4F – Included to address reviewer 1 point 1
3. Figure EV4D – Included to address reviewer 1 point 1
4. Figure EV5A – Included to address reviewer 2's first point
5. Supplemental Table 4 - Included to address reviewer 2's first point

Detailed response to referees:

Referee #1:

The authors have addressed many of my comments adequately and the manuscript is easier to follow now. However, I am still not convinced by the answers to points 4 and 5 regarding the stability of the PI3K 6A mutant complex and its reduced activity in the in vitro assay.

1. The authors argue that the reduced lipid kinase activity of the 6A mutant complex compared to the wild type complex is due to basal phosphorylation of the wt PI3K complex rather than to the reduced stability or purity of the 6A mutant, which however seems evident from the gel shown in Fig. 4E. This hypothesis could easily be tested by incubating the purified wt PI3K complex with a phosphatase. This treatment should reduce its activity to that of the 6A mutant. In addition, it is not comprehensible to this reviewer why the 6E mutant was not included in the assay shown in Fig. 4F.

We thank the referee for their comment. As a result, we have performed a new set of reconstitution assays including VPS34 complexes incorporating VPS15 6SE. The results from these assays were informative and allowed us to significantly clarify our interpretation of the data set as a whole. We also appreciate the logic of the phosphatase experiment suggested; however, any data would be difficult to interpret as the VPS34 complex contains a large number of phosphorylation sites, including multiple ULK substrate residues which likely show a functional degree of basal phosphorylation (Park, 2016 PMID: 27046250). Any dephosphorylation dependent phenotype would be difficult to assign to the 6 sites in question.

Regarding the extra bands evident in the (now updated) Figure 4E, these most likely represent contaminants rather than VPS34 complex components truncated due to instability. Complex I is purified by gel-filtration by collecting the complex I peak fractions, meaning truncated proteins are theoretically excluded. In addition, the 85 kDa

and 70 kDa bands do not comigrate with the complex I in the flotation assays (Figure EV4D). Complex purity varies from batch to batch, largely depending on passage number and viability of the expressing cells. Importantly, the batch used to generate DSF and *in vitro* kinase assay data in the new Figure 4E/F displayed comparable contaminants.

Furthermore, DSF allowed us to accurately quantify the thermal stability of the complexes, and our DSF results suggest that any phenotypic differences are unlikely to be due to instability of the mutant complexes. Supporting this, VPS34 CI incorporating VPS15 6SE phenocopied the 6SA-incorporating complexes in reconstitution assays (see Figure 4E/F), whilst coimmunopurification of VPS34 complex members was unaffected (see Figure EV4B).

Together, when the reconstitution data are considered alongside our rescue data (see Figure 6 and EV5) we believe that prevention of basal phosphorylation of 6S mutant complexes is the most likely driver of the phosphomutant phenotype, although we concede that this is difficult to prove definitively.

2. On page 10 the authors write "The chronic PI3P depletion phenotype observed in CRISPR clones sgA-A8, sgA-B3, sgA-D6 and sgA-D11 confirmed that they are effective VPS15 knockouts (now referred to as VPS15 KOs)". However, I understand that some VPS15 expression still exists in these cells as complete knock out is likely to be lethal. The authors should therefore rephrase this sentence as it could easily be misunderstood.

We have rephrased this to avoid confusion.

3. As a more minor point, the authors write on page 6 that ULK1 would preferentially phosphorylate interacting substrates citing a review (REF 20). This sentence should be clarified as evidently all substrates must at least transiently interact. In addition, rather than a review, the actual evidence should be cited.

We are referring to proteins which stably interact with ULK and will clarify this. We refer here to one of our earlier reviews, specifically to Table 1 in Mercer et al., 2018 (PMID: 29371398). Here we list a multitude of phosphoacceptor residues coming from >30 substrate proteins. Of these we indicate which have been shown to stably bind ULK (>80%). As we include all of the many relevant references there, we didn't incorporate them into this manuscript given that the already large number of references was previously raised by Referee 3 as an issue.

4. On page 11, last sentence the word "showed" or "revealed" should be deleted.

Well spotted, thank you!

Referee #2:

Despite multiple improvements, there are still some issues that should be clarified.

The added data in Fig. 6C would support the authors' claim that phosphorylation of S861 is inhibitory for autophagic flux were it not for the contradictory finding that S861A inhibits flux (Fig. 6B). The authors discuss that "whilst the 6SA rescue/reconstitution data indicate that the ULK-dependent VPS15 phosphorylation positively regulates autophagy, phosphorylation of S861 alone is largely inhibitory" without showing data and also not taking into account the data of 6SA and 6SE VPS15 mutants both showing inhibitory effects (Fig. 4D). As long as both SA and SE mutants demonstrate similar effects, it's difficult to tell how these phosphorylations may regulate autophagy. Determining the precise physiological function of each phosphorylation site may be beyond the scope of this study but determining whether these phosphorylations inhibit or stimulate autophagy is crucial. If S861 is a primary inhibitory site, the authors should at least test the effects of S5E and S5A mutants (with intact S861) to dissect the role of S861 phosphorylation from those of other phosphorylations.

We thank the referee for their helpful comments and have performed the experiment suggested. This has significantly improved the manuscript by a) clarifying the relative contributions of S861 and the other 5 substrate residues to the phosphomutant phenotypes and b) allowing a simple and compelling interpretation of the SA and SE phenotypes.

We have incorporated these changes into the text, but in brief we show that mutation of serine 861 is the main factor in determining the phenotype of 6SA/E-expressing cells. We further suggest that the similarity between the S861A and S861E phenotypes reflects one of two possibilities, either that cyclical phosphorylation and dephosphorylation must occur to drive the phenotype, or that glutamate is an insufficient phosphomimetic in this instance. In both cases, cells expressing VPS15 S861A or S861E would result in a 'un-phosphorylated' phenotype. Accordingly, the most parsimonious interpretation of our data suggests that ULK-dependent phosphorylation of VPS15, primarily at serine 861, promotes autophagy.

The authors added new data demonstrating that FIP200-positive structures accumulate in siATG14 cells in a PI3K CI-dependent manner. However, it has been shown that Vps34 kinase activity regulates the dynamics of the ULK1 complex (e.g., in Ref #1, 21), which may reduce the novelty of the authors' new data. The authors should clarify the difference between these previous studies and the current study.

We have clarified the novelty of our findings on page 17. Briefly, our data are fundamentally different to those reported by Karanasios et al. (ref #1), as they show how preventing VPS34 activity inhibits recruitment of ULK complex proteins to sites of autophagosome formation. In contrast, we show VPS34 inhibition promotes ULK complex recruitment to ubiquitin condensates which are known to nucleate autophagosomes (PMID: 33207181; PMID: 33397898). Our model also allowed us to build on those of Kishi-Itakura et al. (ref #21), as we were able to modulate the recruitment and distribution of autophagy proteins on and within ubiquitin-positive bodies by demonstrating that their structure and constituents are highly sensitive to ULK activity status. We also propose these ubiquitin-positive bodies are hubs of enhanced ULK kinase activity. These findings are of particular interest as preventing the autophagic clearance of similar bodies in the neurons has recently been shown to lead to disease (PMID: 33207181).

Referee #3:

In the revised version of the manuscript, the authors addressed all the points raised by the referees. Moreover, they edited the text that now is more compact and easier to follow.

This referee doesn't have other major concerns regarding the manuscript. Maybe just an advice after reading the rebuttal letter. Page 7 of rebuttal letter, the authors wrote "Whilst phosphorylation of the understanding the biology of the VPS34 complex". Considering this sentence the author may include in the title a mention to VPS34 complex.

Thank you for your comments and for the suggestion regarding the title change, we have changed the title accordingly.

2nd Revision - Editorial Decision

23rd Mar 2021

Thank you for submitting your revised study. The manuscript has now been sent back to referee #1 and #2, whose comments are appended below.

As you will see, the referees state that their remaining criticisms have been sufficiently addressed and recommend the work for publication.

Please find below a list of editorial issues concerning the text and the figures that I need you to address before we can officially accept your manuscript.

Referee #1:

The authors have provided answers to my points and I have no further comments.

Referee #2:

The authors have responded appropriately to my previous comments.

3rd Revision - Editorial Decision

21st Apr 2021

I am pleased to inform you that your manuscript has been accepted for publication in The EMBO Journal.

YOU MUST COMPLETE ALL CELLS WITH A PINK BACKGROUND ↓
PLEASE NOTE THAT THIS CHECKLIST WILL BE PUBLISHED ALONGSIDE YOUR PAPER

Corresponding Author Name: Sharon A Tooze

Journal Submitted to: The EMBO Journal

Manuscript Number: EMBO 2020 105 895